# A Cryptochrome adopts distinct moon- and sunlight states and functions as sun- versus moonlight interpreter in monthly oscillator entrainment

Birgit Poehn[1,2,8], Shruthi Krishnan[3,4,8], Martin Zurl [1,2], Aida Coric[1,2,9], Dunja Rokvic[1,2,9], N. Sören Häfker [1,2], Elmar Jaenicke [3], Enrique Arboleda [1,2,7], Lukas Orel[1,2], Florian Raible[1,2], Eva Wolf [3,4] ✉ & Kristin Tessmar-Raible [1,2,5,6] ✉

The moon's monthly cycle synchronizes reproduction in countless marine organisms. The mass-spawning bristle worm *Platynereis dumerilii* uses an endogenous monthly oscillator set by full moon to phase reproduction to specific days. But how do organisms recognize specific moon phases? We uncover that the light receptor L-Cryptochrome (L-Cry) discriminates between different moonlight durations, as well as between sun- and moonlight. A biochemical characterization of purified L-Cry protein, exposed to naturalistic sun- or moonlight, reveals the formation of distinct sun- and moonlight states characterized by different photoreduction- and recovery kinetics of L-Cry's co-factor Flavin Adenine Dinucleotide. In *Platynereis*, L-Cry's sun- versus moonlight states correlate with distinct subcellular localizations, indicating different signaling. In contrast, r-Opsin1, the most abundant ocular opsin, is not required for monthly oscillator entrainment. Our work reveals a photo-ecological concept for natural light interpretation involving a "valence interpreter" that provides entraining photoreceptor(s) with light source and moon phase information.

Lunar influences on animals are especially well documented in the marine environment[1–3]. Starting with the early twentieth century, numerous scientific studies have shown that the reproductive behavior and sexual maturation of animals as diverse as corals, polychaetes, echinoderms, fishes or turtles are synchronized by the lunar cycle[1,3–7]. In addition, lunar timing effects have also been documented outside the marine environment[8,9], and recently uncovered correlations of human sleep and menstrual cycle properties with moon phases have

re-initiated the discussion of an impact of the moon even on human biology[10,11]. As recently documented for corals, desynchronization of these reproductively critical rhythms by anthropogenic impacts poses a threat to species survival[12].

Despite the importance and widespread occurrence of lunar rhythms, functional mechanistic insight is lacking. Importantly, this synchronization among conspecifics is in many cases not simply a direct reaction to a stimulus, but instead governed by endogenous

[1]Max Perutz Labs, University of Vienna, Vienna BioCenter, Vienna, Austria. [2]Research Platform "Rhythms of Life", University of Vienna, Vienna BioCenter, Dr. Bohr-Gasse 9/4, A-1030 Vienna, Austria. [3]Institute of Molecular Physiology (IMP), Johannes Gutenberg-University, Hanns-Dieter-Hüsch-Weg 17, 55128 Mainz, Germany. [4]Institute of Molecular Biology (IMB), Ackermannweg 4, 55128 Mainz, Germany. [5]Alfred Wegener Institute, Helmholtz Centre for Polar and Marine Research, Am Handelshafen 12, 27570 Bremerhaven, Germany. [6]Carl-von-Ossietzky University, Carl-von-Ossietzky-Straße 9-11, 26111 Oldenburg, Germany. [7]Present address: Institut de Génomique Fonctionnelle de Lyon (IGFL), École Normale Supérieure de Lyon, 32 avenue Tony Garnier, 69007 Lyon, France. [8]These authors contributed equally: Birgit Poehn, Shruthi Krishnan. [9]These authors contributed equally: Aida Coric, Dunja Rokvic. ✉e-mail: evawolf1@uni-mainz.de; kristin.tessmar@mfpl.ac.at

monthly oscillators: circalunar clocks[3,9,13–16]. The marine bristle worm *Platynereis dumerilii* is well documented to possess such a circalunar clock, which controls its reproductive timing and can be entrained by nocturnal light in the lab[5,15,17].

Several reports have linked the expression of cryptochromes (CRYs) with moon phase, suggesting that these genes could be involved in circalunar time-keeping[18], possibly- as proposed for corals- as lunar light receptors[2,19,20]. However, no functional molecular support for such an involvement exists. *P. dumerilii* possesses single-copy gene encoded putative CPD-, (6-4)- and CRY-DASH photolyases, as well as three Cryptochromes[21]. Of those, one (*Pdu*-tr-Cry) is a direct ortholog to the CRYs which function as transcriptional repressors and not light receptors in mammals (termed mammalian CRY1 and mammalian CRY2), as well as in insects (termed CRY2, vCRY or trCRY)[15,21]. One (*Pdu*-L-Cry) is a 1:1 ortholog of a group of Cryptochromes (termed CRY1, dCRY or iCRY), whose founding member *Drosophila melanogaster* dCRY prominently functions as a light receptor in flies[22,23]. First uncovered for insects other than *Drosophila*, many invertebrates possess CRYs that belong to these two principal classes[21,24,25]. Based on the function of their best-characterized members (dCRY and mouse CRY1,2), two cell culture assays are frequently used to test for the possible light receptive and/or transcriptional repressive functions of other (uncharacterized) animal Cryptochrome[24,25]. Using these well-established cell culture assays, we have previously shown that *Pdu*-L-Cry, but not *Pdu*-tr-Cry, undergoes light-dependent degradation, in analogy to dCRY, indicative of a light-receptive function of L-Cry. In contrast, *Pdu*-tr-Cry, but not *Pdu*-L-Cry, showed transcriptional repressor activity (under darkness). Hence we concluded that *Pdu*-tr-Cry would likely function as a transcriptional repressor, while *Pdu*-L-Cry could be light receptive[15]. As informative data are entirely lacking for plant-like pt-CRYs in animals (the third type of CRY that *Platynereis* possesses[21]), we decided to focus on the Cryptochrome that showed the most likely evidence for a possible light-receptor function[15] and investigated the functional role and biochemical properties of the light-receptive Cryptochrome L-Cry in the annelid *P. dumerilii*.

In order for organisms to synchronize their monthly oscillator they need to be entrained to a specific moon phase. However, this requires that the entraining mechanisms can discriminate between naturalistic sun- and moonlight, as well as the different moon phases. The latter differ predictably by moonlight intensity and the duration for which the moon is present on the night sky[26]. Classical and recent work on the monthly oscillators of the midge *Clunio* and the bristle worm *Platynereis* established that mimicking the duration of full moon (i.e. continuous light at night) is sufficient to entrain their monthly oscillators[5,13,15,27]. We thus followed and subsequently expanded on these established experimental paradigms and focussed on new moon versus full moon conditions throughout the experiments.

In this work, prompted by the prediction from corals[2] and our own previous study suggesting that *Pdu*-L-Cry might function as a light receptor in a heterologous cell expression system[15], we studied genetic loss-of-function mutants of *Platynereis l-cry* and combined these investigations with biochemical and in vivo cell biological analyses using custom-designed light sources based on long-term measurements of light intensity and spectra at the worms' natural habitat. We revealed that L-Cry is able to discriminate between different moonlight durations, as well as between sun- and moonlight. Consistent with L-Cry's function as light interpreter its genetic loss led to a faster re-entrainment under artificially strong nocturnal light, but had only very minor effects under naturalistic sun- and moonlight compared to wild type. This suggested that L-Cry was blocking the "wrong" light from impacting on the monthly oscillator. In contrast, r-Opsin1, the most abundant ocular opsin, was not required for monthly oscillator entrainment. The biochemical characterization of purified L-Cry protein revealed the formation of distinct dark, sun- and moonlight states that can be explained by different quantum efficiencies for the FAD

photoreduction responses in the two monomers of the L-Cry dimer. The formation of the distinct sunlight versus moonlight biochemical states correlated with distinct subcellular localizations, indicative of differential signaling of these states within the cells of the worm. These findings advance our molecular mechanistic understanding of a fundamental biological phenomenon: moon-controlled monthly timing. Such level of understanding is also an essential prerequisite to tackle anthropogenic threats on marine ecology.

## Results

### *l-cry* mutants show higher spawning synchrony than wild-type animals under non-natural light conditions

In order to test for a functional involvement of L-Cry in monthly oscillator function, we generated two *l-cry* mutant alleles (Δ34 and Δ11bp) (Fig. 1a) using TALENs[28]. In parallel, we generated a monoclonal antibody against *Platynereis* L-Cry. By testing mutant versus wildtype worms with the anti-L-Cry antibody in Western blots (Fig. 1b) and immunohistochemistry (Fig. 1e–j), we verified the absence of L-Cry protein in mutants. Furthermore, we confirmed that the staining of the antibody in wildtype worms (Fig. 1e–h) matches the regions where *l-cry* mRNA is expressed (Fig. 1d). These tests confirmed that the engineered *l-cry* mutations result in loss-of-function alleles. In turn, they validate the specificity of the raised anti-L-Cry antibody.

We next assessed the circalunar maturation timing of wild types and *l-cry* mutant populations in conventional culture conditions, i.e. worms grown under typical indoor room lighting (named here artificial sun- and moonlight, Supplementary Fig. 1b).

We expected either no phenotype (if L-Cry was not involved in circalunar clock entrainment) or a decreased spawning precision (if L-Cry was functioning as moonlight receptor in circalunar clock entrainment). Instead we observed an increased precision of the entrained worm population:

We analyzed the maturation data using two statistical approaches, linear and circular statistics. We used the classical linear plots[5] and statistics to compare the monthly spawning data distribution (Fig. 2a–c, i). This revealed a clear difference between mutant animals, which exhibited a stronger spawning peak at the beginning of the NM phase, compared to their wildtype and heterozygous counterparts (Fig. 2a–c, Kolmogorov–Smirnov test on overall data distribution, Fig. 2i).

We then analyzed the same data using circular statistics (as the monthly cycle is repeating, see details in Methods section), which allowed us to describe the data with the mean vector (defined by the direction angle μ and its length r, shown as arrows in Fig. 2e–g). The phase coherence r (ranging from 0 to 1) serves as a measure for synchrony of the population data. As expected for entrained populations, all genotypes distributed their spawning across a lunar month significantly different from random (Fig. 2e–g, p values in circles, Rayleigh's Uniformity test[29]). In line with the observed higher spawning peak of the *l-cry*−/− mutants in the linear plots, the circular analysis revealed a significant difference in spawning distribution (Mardia–Watson-Wheeler test, for details see Methods section) and higher spawning synchrony of mutants (r = 0.614) than in wild types and heterozygotes (r = 0.295 and r = 0.222) (Fig. 2i). The specificity of this phenotype of higher spawning precision for *l-cry* homozygous mutants was confirmed by analyses on trans-heterozygous *l-cry* (Δ34/Δ11) mutants (Supplementary Fig. 2), and by the fact that such a phenotype is not detectable in any other light receptor mutant available in *Platynereis* (r-opsin1[30]: Supplementary Fig. 3a, b, e, f, i; c-opsin1[31]: Supplementary Fig. 3c, d, g, h, i, Go-opsin: refs. 32, 33).

### The higher spawning synchrony of *l-cry* mutants under artificial light mimics the spawning precision of wild-type at its natural habitat

This increased spawning precision of *l-cry* mutants under artificial (but conventional indoor) laboratory light conditions let us wonder about

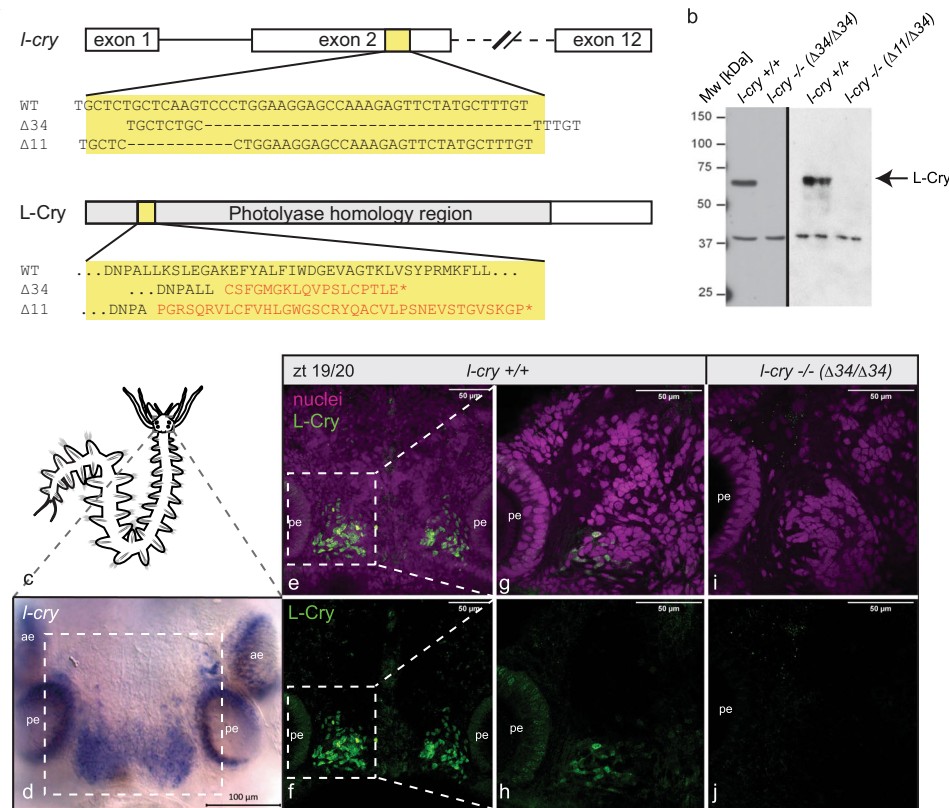

**Fig. 1 | l-cry$^{-/-}$ mutants are loss-of-function alleles. a** Overview of the l-cry genomic locus for wt and mutants. Both mutant alleles result in an early frameshift and premature stop codons. The Δ34 allele has an additional 9 bp deletion in exon 3. **b** Western Blots of *P. dumerilii* heads probed with anti-L-Cry antibody. In the context of further investigations such Western blots of mutant versus wild types have been performed more than 10 times with highly consistent results. Also see further analyses in this manuscript and ref. 36. **c** overview of *P. dumerilii*. **d** whole mount in situ hybridization against l-cry mRNA on worm head. ae, anterior eye; pe, posterior eye. **e–j** Immunohistochemistry of premature wild-type (**e–h**) and mutant (**i, j**) worm heads sampled at zt19/20 using anti-L-Cry antibody (green) and Hoechst staining (magenta), dorsal views, anterior up. **e, f**: z-stack images (maximal projections of 50 layers, 1.28 µm each) in the area highlighted by the rectangle in (**d**), whereas (**g–j**) are single layer images of the area highlighted by the white rectangles in (**e, f**). In the context of further investigations such stainings of mutant versus wild types have been performed more than 10 times with highly consistent results. Also see further analyses in this manuscript and ref. 36.

the actual population synchrony of the worms under truly natural conditions. The lunar spawning synchrony of *P. dumerilii* at the Bay of Naples (the origin of our lab culture) has been worked on for more than 100 y. This allowed us to re-investigate very detailed spawning data records from the worms' natural habitat published prior to environmental/light pollution. For better accessibility and comparability we combined all months and replotted the data published in 1929[34] (Fig. 2d, h, I; see details in Methods section; $r = 0.631$). This analysis revealed that the higher spawning synchrony in *l-cry$^{-/-}$* worms mimics the actual spawning synchrony of *P. dumerilii* populations in their natural habitat[34] (compare Fig. 2c, g with 2d, h.)

Given that recent, non-inbred isolates from the same habitat as our lab inbred strains (which is the same habitat as the data collected in ref. 34) exhibit a broad spawning distribution under standard worm culture light conditions (which includes the bright artificial moonlight)[35], we hypothesized that the difference in spawning synchrony between wildtype laboratory cultures and populations in their natural habitat is caused by the rather bright nocturnal light stimulus typically used for the standard laboratory culture (Supplementary Fig. 1a vs. b).

### Lunar spawning precision of wild-type animals depends on naturalistic moonlight conditions

We next tested the resulting prediction that naturalistic moonlight should increase the spawning precision of the wildtype population, using naturalistic sun- and moonlight devices we specifically designed based on light measurements at the natural habitat of *P. dumerilii*[31]

(Supplementary Fig. 1a, c). We assessed the impact of the naturalistic sun- and moonlight (Supplementary Fig. 1a, c) on wildtype animals, maintaining the temporal aspects of the lab light regime (i.e. 8 nights of "full moon"). Indeed, merely adjusting the light intensity to naturalistic conditions increased the precision and phase coherence of population-wide reproduction: After several months under naturalistic sun- and moonlight, wildtype worms spawned with a major peak highly comparable to the wildtype precision reported at its natural habitat (Fig. 2d, h vs. j, k), and also exhibited an increased population synchrony ($r = 0.398$ compared to $r = 0.295$ under standard worm room light conditions). This increased similarity to the spawning distribution at the natural habitat ("Sea") is confirmed by statistical analyses (Fig. 2l): The phase difference (angle between the two mean vectors) is only one day (corresponding to 12°). In contrast, the spawning distribution of wild types under standard worm room light versus naturalistic light conditions is highly significantly different in linear and circular statistical tests and has a phase difference of 7.7 days (Fig. 2l).

These findings show that it is the naturalistic light that is critical for a highly precise entrainment of the monthly clock of wild-type worms. Given that *l-cry$^{-/-}$* animals reach this high precision with the artificial light (i.e. standard lab light) implies that in wildtype L-Cry blocks artificial, but not naturalistic full-moonlight from efficiently synchronizing the circalunar clock. This block is removed in *l-cry$^{-/-}$* animals, leading to a better synchronization of the *l-cry$^{-/-}$* population. This finding suggests that L-Cry's major role could be that of a gatekeeper controlling which ambient light is interpreted as full-moonlight stimulus for circalunar clock entrainment.

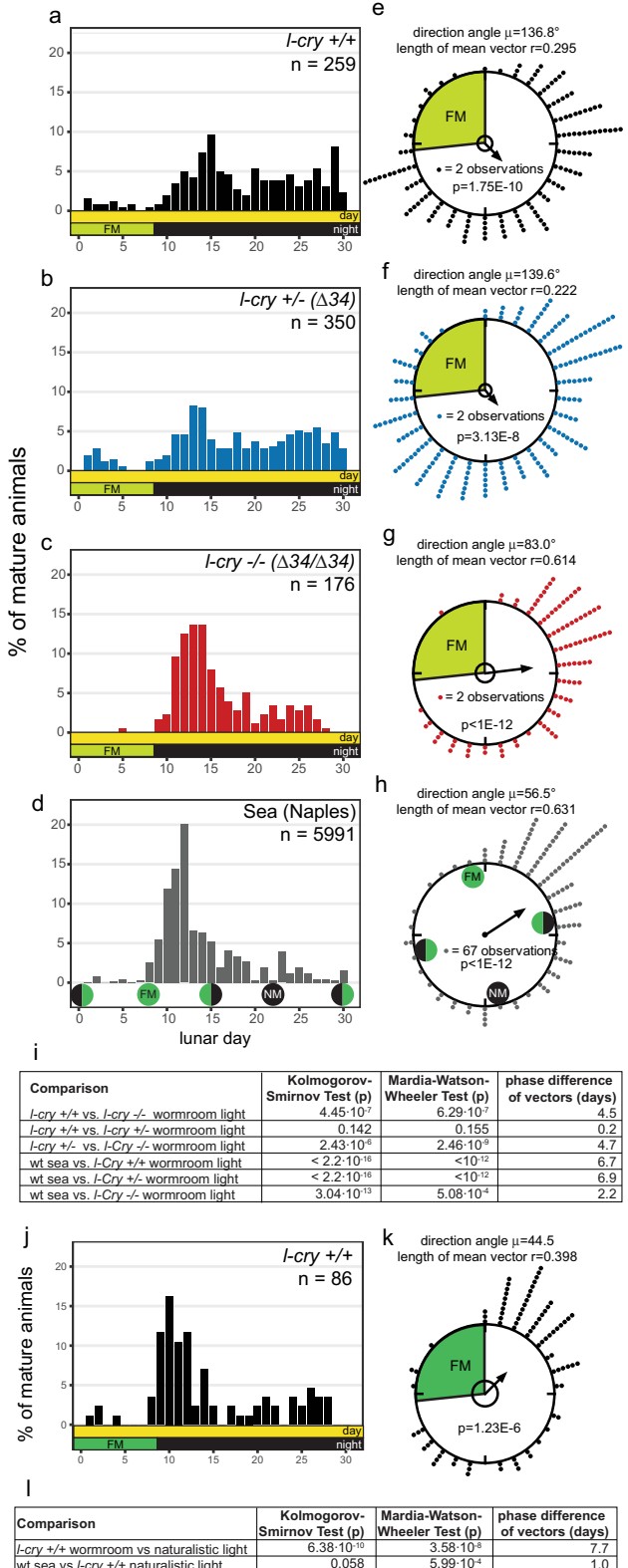

**Fig. 2 | L-Cry shields the circalunar clock from light that is not naturalistic moonlight. a–d, j** Spawning of *l-cry* ⁺/⁺ (**a**), *l-cry* ⁺/⁻ ⁽Δ³⁴⁾ (**b**) and *l-cry* ⁻/⁻⁽Δ³⁴/Δ³⁴⁾ (**c**) animals over the lunar month in the lab with 8 nights of artificial moonlight (**a–c**), under natural conditions in the sea (**d**, replotted from ref. 34,50,) and in the lab using naturalistic sun- and moonlight (j, 8 nights moonlight). **e–h, k** Data as in (**a–d, j**) as circular plot. 360° correspond to 30 days of the lunar month. The arrow represents the mean vector, characterized by the direction angle μ and r (length of μ). r indicates phase coherence (measure of population synchrony). *p*-values inside the plots: result of Rayleigh Tests. Significance indicates non-random distribution of data points. The inner circle represents the Rayleigh critical value (*p* = 0.05). **i–l** Results of two-sided multisample statistics on spawning data shown in (**a–h, j, k**). The phase differences in days can be calculated from the angle between the two mean vectors (i.e. 12°= 1 day).

We thus compared the spawning rhythms of *l-cry*⁺/⁺ and *l-cry*⁻/⁻ worms under a re-entrainment paradigm, where we provided our bright artificial culture full-moonlight at the time of the subjective new moon phase (Fig. 3a). In order to compare the spawning data distribution relative to the initial full moon (FM) stimulus, as well as to the new full moon stimulus (i.e. new FM), we used two nomenclatures for the months: months with numbers are analyzed relative to the initial nocturnal light stimulus (i.e. FM), whereas months with letters are analyzed relative to the new (phase-shifted) nocturnal light stimulus (i.e. new FM, Fig. 3a). When the nocturnal light stimulus is omitted (to test for the oscillator function) we then refer to 'free-running FM' (FR-FM) or 'new free-running FM' (new FR-FM), respectively (Fig. 3a). Using these definitions, the efficiency of circalunar clock re-entrainment will be reflected in the similarity of spawning data distributions between month 1 and month D, i.e. the more similar the distribution, the more the population has shifted to the new phase.

When using the artificial nocturnal light conditions, the re-entrainment of *l-cry*⁻/⁻ animals was both faster and more complete than for their wildtype relatives, as predicted from our gate keeper hypothesis. This is evident from the linear data analysis and Kolmogorov−Smirnov tests when comparing the month before the entrainment (month 1) with two months that should be shifted after the entrainment (months C,D, Fig. 3b, c, f, g).

Most notably, while *l-cry*−/− worms were fully shifted in month D (Fig. 3c: compare boxes and see complete lack of spawning at the light blue arrowhead indicating the old NM/new FR-FM phase versus massive spawning at new NM phase around dark blue arrowhead), wildtype animals were still mostly spawning according to the initial lunar phase (Fig. 3b: compare boxes and see spawning at the light blue arrowhead versus almost lack of spawning at dark blue arrowhead). The faster re-entrainment of *l-cry*−/−, compared to *l-cry*+/+ animals is also confirmed by the Mardia−Watson-Wheeler test (see Methods section for details). For *l-cry*⁺/⁺ animals, the comparisons of the spawning distributions before and after re-entrainment show a 1000-fold (months 1 versus C) and tenfold (months 1 versus D) higher statistical significance difference than the corresponding comparisons for *l-cry*−/− worms (Fig. 3f, g). Consistently, the phase differences in days calculated from the angle between the two mean vectors from the circular analysis is smaller in the mutants than in the wild types when comparing the phase of the month before the entrainment (month 1) with two months after the entrainment (months C, D) (Fig. 3d–g). The fact that there are still differences in the mutant population before and after entrainment is likely due to the fact that even the mutants are not fully re-entrained. However, they have shifted more robustly in response to an artificial nocturnal light stimulus than the wild types. This provides further evidence that in wildtype worms L-Cry indeed blocks the "wrong" light from entering into the circalunar clock and thus functions as a light gatekeeper.

### *l-cry* functions as a light signal gatekeeper for circalunar clock entrainment

A prediction of this hypothesis is that mutants should entrain better to an artificial full-moonlight stimulus provided out-of-phase than their wild type counterparts (in which L-Cry should block the "wrong" moonlight at least partially from re-entraining the circalunar oscillator).

| Comparison | Kolmogorov-Smirnov Test (p) | Mardia-Watson-Wheeler Test (p) | phase difference of vectors (days) |
|---|---|---|---|
| *l-cry* +/+ vs. *l-cry* -/-  wormroom light | 4.45·10⁻⁷ | 6.29·10⁻⁷ | 4.5 |
| *l-cry* +/+ vs. *l-cry* +/- wormroom light | 0.142 | 0.155 | 0.2 |
| *l-cry* +/-  vs. *l-Cry* -/- wormroom light | 2.43·10⁻⁶ | 2.46·10⁻⁹ | 4.7 |
| wt sea vs. *l-Cry* +/+ wormroom light | < 2.2·10⁻¹⁶ | <10⁻¹² | 6.7 |
| wt sea vs. *l-Cry* +/- wormroom light | < 2.2·10⁻¹⁶ | <10⁻¹² | 6.9 |
| wt sea vs. *l-Cry* -/- wormroom light | 3.04·10⁻¹³ | 5.08·10⁻⁴ | 2.2 |

| Comparison | Kolmogorov-Smirnov Test (p) | Mardia-Watson-Wheeler Test (p) | phase difference of vectors (days) |
|---|---|---|---|
| *l-cry* +/+ wormroom vs naturalistic light | 6.38·10⁻¹⁰ | 3.58·10⁻⁸ | 7.7 |
| wt sea vs *l-cry* +/+ naturalistic light | 0.058 | 5.99·10⁻⁴ | 1.0 |

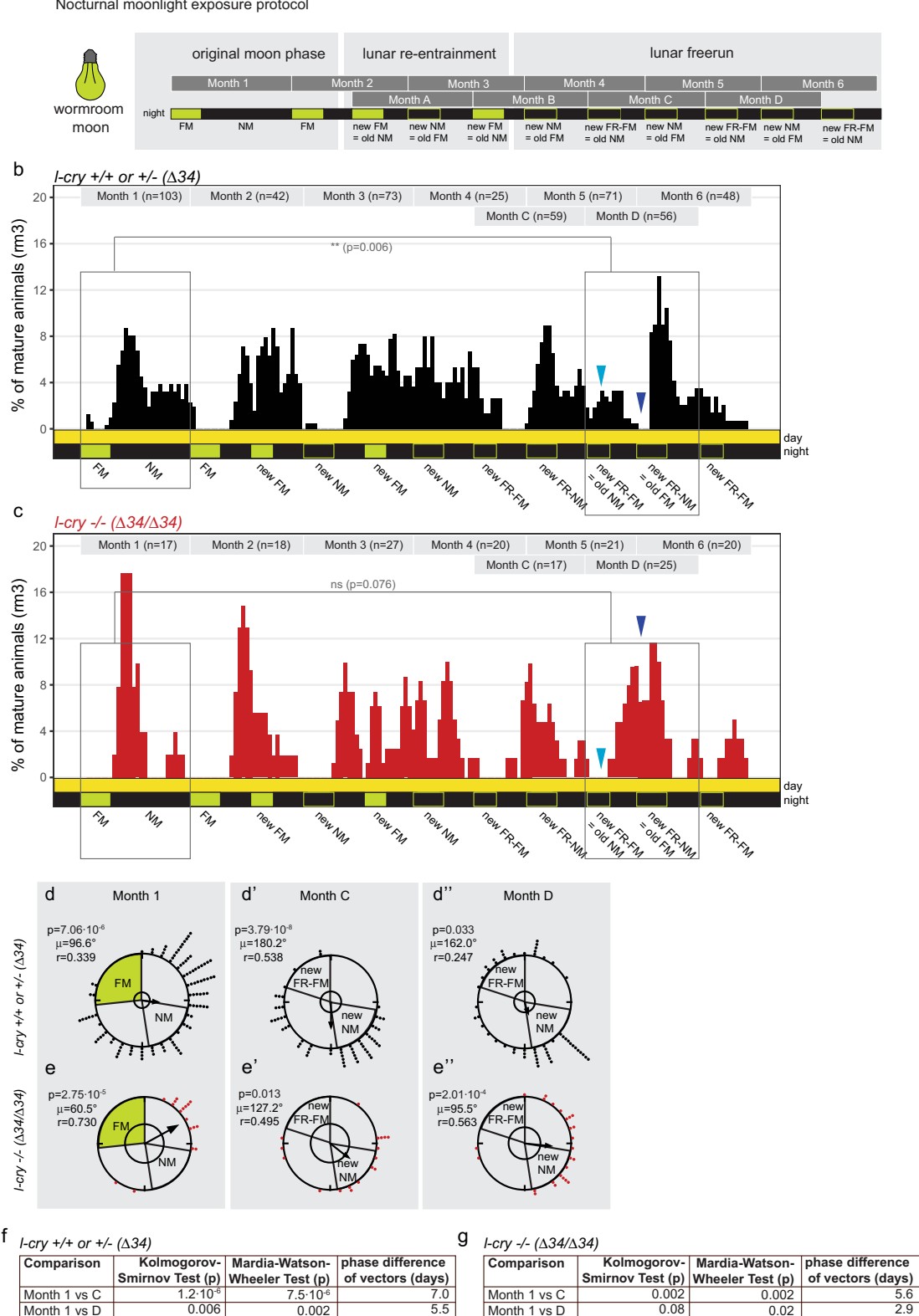

**L-Cry functions mainly as light interpreter, while its contribution as direct moonlight entraining photoreceptor is (at best) minor**

We next tested to which extent L-Cry is itself a sensor for the re-entrainment signal under naturalistic light conditions. Based on the finding that *l-cry−/−* worms can still re-entrain the circalunar oscillator (see above), it is clear that even if L-Cry also directly contributed to the entrainment, it cannot be the only moonlight receptor mediating entrainment. With the experiments below, we aimed to test if L-Cry has any role as an entraining photoreceptor to the monthly oscillator.

Thus, we tested how the circalunar clock is shifted in response to a re-entrainment with naturalistic moonlight in *Platynereis* wt versus *l-cry−/−* worms. For this, animals initially raised and entrained under standard worm room light conditions of artificial sun- and moonlight

**Fig. 3 | *l-cry−/−* mutants entrain the circalunar clock faster than wt to a high-intensity artificial moonlight stimulus. a** Nocturnal moonlight exposure protocol of lunar phase shift (entrained by 8 nights, phased shifted by 6 nights of artificial culture moon, light green). **b, c** Number of mature animals (percent per month, rolling mean with a window of 3 days) of *l-cry* wild-type (**b**) and homozygous mutant (**c**) animals. *p*-values indicate results of Kolomogorov–Smirnov tests. Dark blue arrowheads- old FM phase: wt show a spawning minimum, indicative that the worms are not properly phase shifted. Mutants spawn in high numbers, but don't spawn at the old NM indicated by light blue arrowhead. Also compare to initial FM and NM in months 1, 2. **d, e** Circular plots of the data shown in (**b**) and (**c**). Each circle represents one lunar month. Each dot represents one mature worm. The arrow represents the mean vector characterized by the direction angle μ and r. r (length of μ) indicates phase coherence (measure of population synchrony). The inner circle represents the Rayleigh critical value (*p* = 0.05). **f, g** Results of two-sided multi-sample statistics of data in (**d, e**). Phase differences in days can be calculated from the angle between the two mean vectors (i.e. 12° = 1 day).

(Supplementary Fig. 1b, e) were challenged by a deviating FM stimulus of 8 nights of naturalistic moonlight (Fig. 4a, Supplementary Fig. 1c, e). This re-entraining stimulus was repeated for three consecutive months (Fig. 4a).

The resulting spawning distribution was analyzed for the efficacy of the naturalistic moonlight to phase-shift the circalunar oscillator. In order to test if the animals had shifted their spawning to the new phase, we again compared the spawning pattern before the exposure to the new full moon stimulus (months with numbers: data distribution analyzed relative to the initial/old FM, see Fig. 4a for an overview) to the spawning pattern after the exposure to the new full moon stimulus (months with letters: data distribution analyzed relative to the new FM, Fig. 4a). The more similar the data distributions of month 1 is to the months C, D, the more the population was shifted to the new phase.

The first re-entraining full moon stimulus (Fig. 4b, first dark green box) is given in the middle of the main spawning period. The nocturnal light itself does not cause immediate effects on the number of spawning worms (Fig. 4b, see also Fig. 2b, c), but the repeated exposure resulted in a noticeable shift of the spawning distribution indicating a phase shift of the monthly oscillator in wildtype. Already at the third re-entraining full moon stimulus, wildtype animals exhibited a completely shifted spawning pattern (Fig. 4b, d-d″, month 1, 2 vs. month C). This is supported by statistical analyses: When comparing the months 1 and 2 (relative to the old FM before the shift) to the month C (relative to the new FM after the shift), both the Kolmogorov–Smirnov test (Fig. 4b: gray rectangles, 4f) and the Mardia–Watson–Wheeler test of the same data were non-significant (Fig. 4f), indicative of the population shifting to the new phase. Consistently, the direction angle (μ) of the mean vectors before and after the shift was highly similar, resulting in a phase difference of only 0.2 days between months 1 and C and 0.5 days between month 2 and month C (Fig. 4f, for details see methods). The month under circalunar free-running conditions (month D) supports this observation, albeit with lower statistical support (Fig. 4b, d″, f).

Of note, wild-type worms would eventually reach the high spawning precision found under naturalistic moonlight only after several more months based on independent experiments (Fig. 2j, k).

When we analyzed the spawning distribution of *l-cry* mutants in the same way as the wild types, we found that the data distribution exhibited significant differences in the linear Kolmogorov–Smirnov test when comparing months 1 and 2 before the shift to the months C and D after the shift (Fig. 4c: gray rectangles, Fig. 4g); as well as in the phase distribution in the circular analyses when comparing the months before the shift (months 1 and 2) with the last months of the shift (months C,D) (Fig. 4e, e′ versus e″, e‴, g). The populations also exhibited a noticeable phase difference of ≥3.5 days (Fig. 4g).

Based on the statistical significant difference in the re-entrainment of *l-cry−/−*, but not wild-type populations under a naturalistic sun- and moonlight regime, we conclude that L-Cry also likely contributes to circalunar entrainment as a photoreceptor. However, as these differences are rather minor, compared to the much stronger differences seen under artificial light regime, we conclude that its major role is the light gatekeeping function.

In an independent study that focused on the impact of moonlight on daily timing, we identified r-Opsin1 as a lunar light receptor that mediates moonlight effects on the worms' -24 h clock[36]. We tested if

*r-opsin1* is similarly important for mediating the moonlight effects on the monthly oscillator of the worm, analyzed here. This is not the case. *r-opsin1−/−* animals re-entrain as well as wildtype worms under naturalistic light conditions (Supplementary Fig. 4). This adds to and is also consistent with our above observation that the spawning distribution is un-altered between *r-opsin1−/−* and wildtype animals under artificial light conditions (Supplementary Fig. 3a, b, e, f). This finding also further enforces the notion that monthly and daily oscillators use distinct mechanisms, but both require L-Cry as light interpreter.

## L-Cry discriminates between naturalistic sun- and moonlight by forming differently photoreduced states

Given that the phenotype of *l-cry−/−* animals suggests a role of L-Cry as light gatekeeper, i.e. only allowing the 'right' light to most efficiently impact on the circalunar oscillator, we next investigated how this could function on the biochemical and cell biological level.

While we have previously shown that *Pdu*-L-Cry is degraded upon light exposure in S2 cell culture[15], it has remained unclear if L-Cry has the spectral properties and sensitivity to sense moonlight and whether this would differ from sunlight sensation. To test this, we purified full length L-Cry from insect cells (Supplementary Fig. 5a–c). Multi-angle light scattering (SEC-MALS) analyses of purified dark-state L-Cry revealed a molar mass of about 130 kDa, consistent with the formation of an L-Cry homodimer (theoretical molar mass of L-Cry monomer is 65.6 kDa) (Fig. 5a). Furthermore, purified L-Cry binds Flavin Adenine Dinucleotide (FAD) as its chromophore (Supplementary Fig. 5d, e). We then used UV/Vis absorption spectroscopy to analyze the FAD photoreaction of purified L-Cry in presence of 1 mM TCEP to prevent protein oxidation. The absorption spectrum of dark-state L-Cry showed maxima at 450 nm and 475 nm, consistent with the presence of oxidized FAD (Supplementary Fig. 5f, black line). As basic starting point to analyze its photocycle, L-Cry was photoreduced using a LED (PerkinElmer ACULED Dyo) with a blue-light dominated spectrum and spectral peak at 450 nm (Supplementary Fig. 1d, d′, henceforth referred to as "blue-light") for 110 s[37]. The light-activated spectrum showed that blue-light irradiation of L-Cry leads to the complete conversion of $FAD_{ox}$ into an anionic FAD radical (FAD°⁻) with characteristic FAD°⁻ absorption maxima at 370 nm and 404 nm and reduced absorbance at 450 nm (Supplementary Fig. 5f, blue spectrum, black arrows). In darkness, L-Cry reverted back to the dark-state with time constants of 2 min (18 °C), 4 min (6 °C) and 4.7 min (ice) (Supplementary Fig. 5g–k).

We then investigated the response of L-Cry to ecologically relevant light, i.e. sun- and moonlight using naturalistic sun- and moonlight devices that we designed based on light measurements at the natural habitat of *P. dumerilii*[31] (Supplementary Fig. 1a, c, e). Upon naturalistic sunlight illumination, FAD was photoreduced to FAD°⁻, but with slower kinetics than under the stronger blue-light source, likely due to the intensity differences between the two lights (Supplementary Fig. 1c–e).

While blue-light illumination led to a complete photoreduction within 110 s (Supplementary Fig. 5f), sunlight induced photoreduction to FAD°⁻ was completed after 5–20 min (Fig. 5b) and did not further increase upon continued illumination for up to 2 h (Supplementary Fig. 6a). Dark recovery kinetics had time constants of 3.2 min (18 °C) and 5 min (ice) (Fig. 1c, Supplementary Fig. 6b, c).

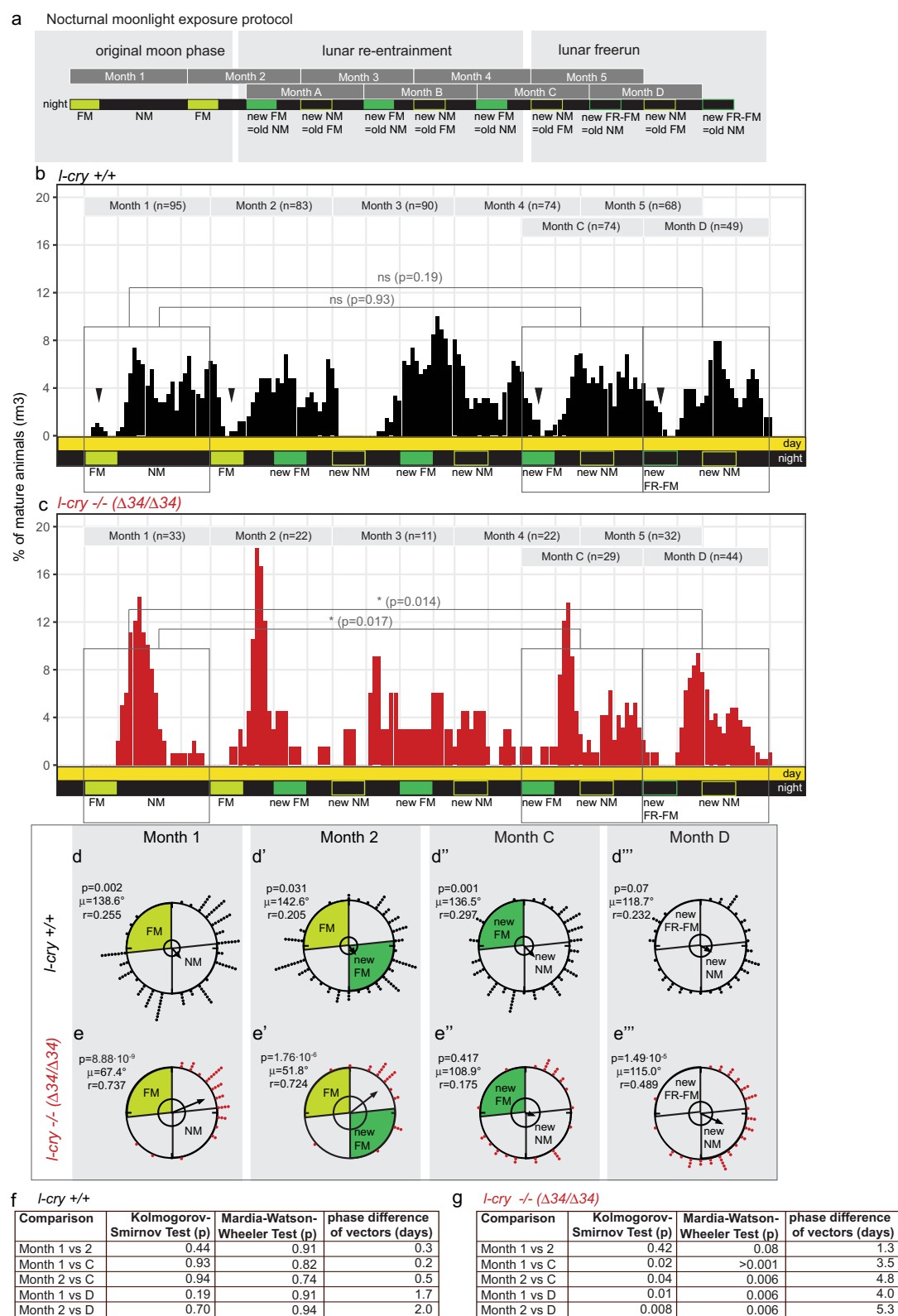

**Fig. 4 | l-cry has a minor contribution as entraining photoreceptor to circalunar clock entrainment. a** Nocturnal moonlight exposure protocol of lunar phase shift with 8 nights of naturalistic moonlight (dark green). Number of mature animals (percent per month, rolling mean with a window of 3 days) of *l-cry* wild-type (**b**) and mutant (**c**) animals. *p*-values: Kolomogorov–Smirnov tests. Black arrowheads indicate spawning-free intervals of the wildtype, which shifted to the position of the new FM (under free-running conditions: FR-FM). **d**, **e** Data as in (**b**, **c**) plotted as

circular data. 360° correspond to 30 days of the lunar month. The arrow represents the mean vector characterized by the direction angle μ and r. r (length of μ) indicates phase coherence (measure of population synchrony). *p* values are results of Rayleigh Tests: Significance indicates non-random distribution of data points. The inner circle represents the Rayleigh critical value (*p* = 0.05). **f**, **g** Results of two-sided multisample statistics on spawning data shown in (**a**–**e**). Phase differences in days can be calculated from the angle between the two mean vectors (i.e. 12°= 1 day).

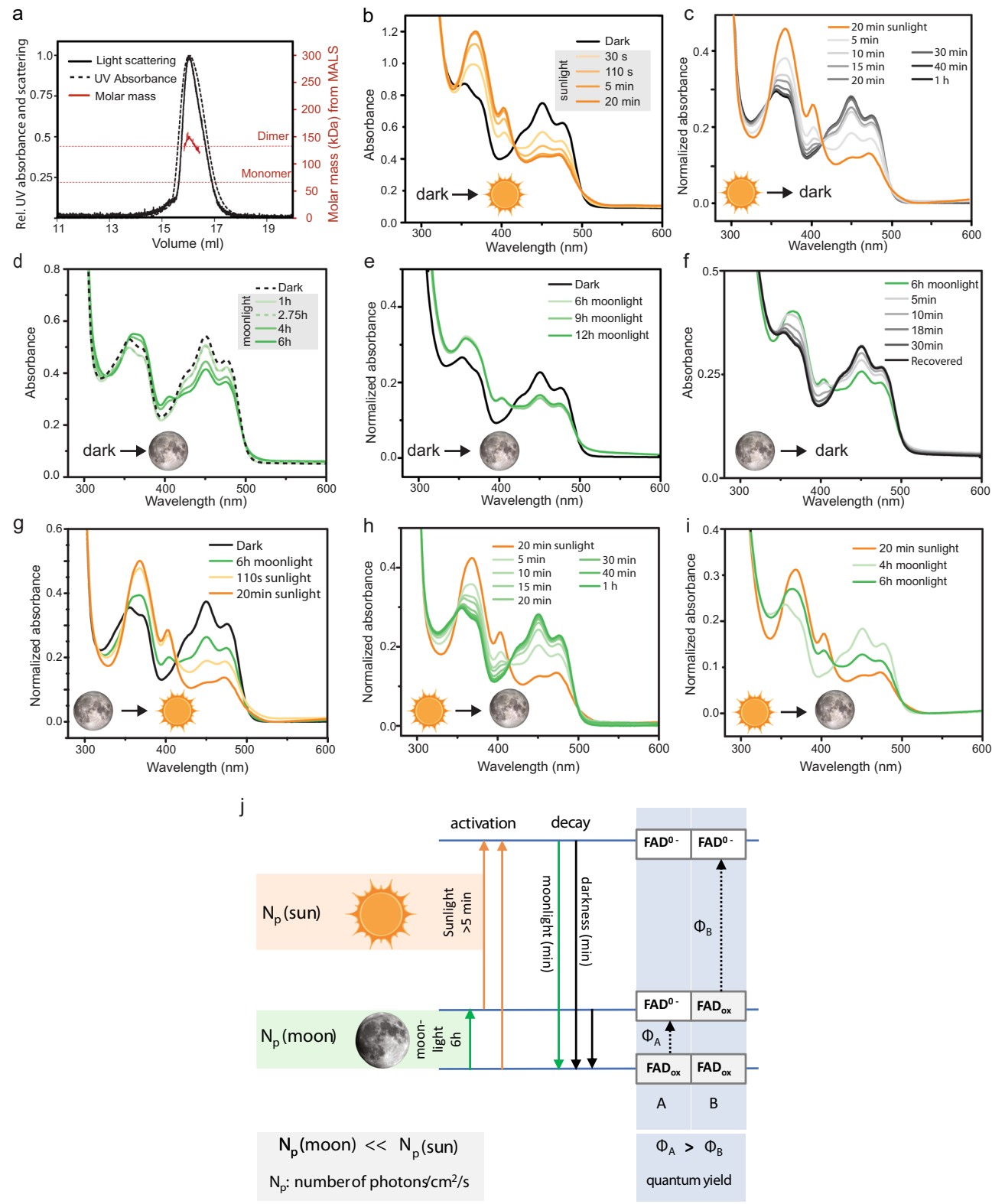

As the absorbance spectrum of L-Cry overlaps with that of moonlight at the *Platynereis* natural habitat (Supplementary Fig. 1a), L-Cry has the principle spectral prerequisite to sense moonlight. However, the most striking characteristic of moonlight is its very low intensity ($5.8 \times 10^{10}$ photons/cm²/s at −5m, Supplementary Fig. 1a–e). To test if *Pdu*-L-Cry is sensitive enough for moonlight, we illuminated purified L-Cry with our custom-built naturalistic moonlight, closely resembling full-moonlight intensity and spectrum at the *Platynereis*

natural habitat (Supplementary Fig. 1a, c, e). Naturalistic moonlight exposure up to 2.75 h did not markedly photoreduce FAD, notably there was no difference between 1 h and 2.75 h (Fig. 5d). However, further continuous naturalistic moonlight illumination of 4 h and longer resulted in significant changes (Fig. 5d), whereby the spectrum transitioned towards the light activated state of FAD°⁻ (note peak changes at 404 nm and at 450 nm). This photoreduction progressed further until 6 h naturalistic moonlight exposure (Fig. 5d). No

**Fig. 5 | L-Cry forms differently photoreduced sunlight- and moonlight states.**
**a** Multi-Angle Light Scattering (MALS) analyses of dark-state L-Cry fractionated by size exclusion chromatography (SEC). Black dashed line: normalized UV absorbance, solid line: normalized scattering signal. The molar mass of about 130 kDa derived from MALS (mass signal shown in red) corresponds to an L-Cry homodimer. **b** Absorption spectrum of L-Cry in darkness (black) and after sunlight exposure (orange). Additional timepoints: Supplementary Fig. 6a. **c** Dark recovery of L-Cry after 20 min sunlight on ice. Absorbance at 450 nm in Supplementary Fig. 6b. **d, e** Absorption spectra of L-Cry after exposure to naturalistic moonlight for different durations. **f** Full spectra of dark recovery after 6 h moonlight. Absorbance

at 450 nm: Supplementary Fig. 6d. **g** Absorption spectrum of L-Cry after 6 h of moonlight followed by 20 min of sunlight. **h** Absorption spectrum of L-Cry after 20 min sunlight followed by moonlight first results in dark-state recovery. Absorbance at 450 nm: Supplementary Fig. 6e. **i** Absorption spectrum of L-Cry after 20 min sunlight followed by 4 h and 6 h moonlight builds up the moonlight state. **j** Model of L-Cry responses to sunlight (orange), moonlight (green) and darkness (black). Only transitions between stably accumulating states are shown. Absorbances in (b–i) were normalized when a shift in the baseline occurred between different measurements of the same measurement set, which is then indicated on the Y-axis as "normalized absorbance".

additional photoreduction could be observed after 9 h and 12 h of naturalistic moonlight exposure (Fig. 5e), indicating a distinct state induced by naturalistic moonlight that reaches its maximum after ~6 h, when about half of the L-Cry molecules are photoreduced. This time of ~6 h is remarkably consistent with classical work showing that a minimum of ~6 h of continuous nocturnal light is important for circalunar clock entrainment, irrespective of the preceding photoperiod[5]. The dark recovery of L-Cry after 6 h moonlight exposure occurred with a time constant of 6.7 min at 18 °C (Fig. 5f, Supplementary Fig. 6d). Given that both sunlight and moonlight cause FAD photoreduction, but with different kinetics and different final $FAD^{o-}$ product/$FAD_{ox}$ educt ratios, we wondered how purified L-Cry would react to transitions between naturalistic sun- and moonlight (i.e. during "sunrise" and "sunset").

Mimicking the sunrise scenario, L-Cry was first illuminated with naturalistic moonlight for 6 h followed by 20 min of sunlight exposure. This resulted in an immediate enrichment of the $FAD^{o-}$ state (Fig. 5g). Hence, naturalistic sunlight immediately photoreduces remaining oxidized flavin molecules, that are characteristic of moonlight activated L-Cry, to $FAD^{o-}$, to reach a distinct fully reduced sunlight state.

In contrast, when we next mimicked the day-night transition ("sunset") by first photoreducing with naturalistic sunlight (or strong blue-light) and subsequently exposed L-Cry to moonlight, L-Cry first returned to its full dark-state within about 30 min (naturalistic sunlight: $\tau = 7$ min (ice), Fig. 5h, Supplementary Fig. 6e; blue-light: $\tau = 9$ min (ice), Supplementary Fig. 6f–h), despite the continuous naturalistic moonlight illumination. Prolonged moonlight illumination then led to the conversion of dark-state L-Cry to the moonlight state (Fig. 5i, Supplementary Fig. 6f). Hence, fully photoreduced sunlight-state L-Cry first has to return to the dark-state before accumulating the moonlight state characterized by the stable presence of the partial $FAD^{o-}$ product/$FAD_{ox}$ educt. In contrast to sunlight-state L-Cry, moonlight-state L-Cry does not return to the oxidized (dark) state under naturalistic moonlight (Fig. 5e), i.e. moonlight maintains the moonlight state, but not the sunlight state. We note, that a partially photoreduced L-Cry state may be formed transiently during dark-state recovery of the sunlight state under moonlight. However, this transiently occurring partially photoreduced L-Cry state would differ from the "true" moonlight state (e.g. by an allosteric change) preventing its accumulation (see discussion and Supplementary Fig. 6i).

Given that L-Cry forms a homodimer and moonlight photoreduces about half of the FAD molecules, we propose that the moonlight state corresponds to a half-reduced $FAD^{o-}$ $FAD_{ox}$ dimer, where FAD is only photoreduced in one L-Cry monomer, whereas in the sunlight state both monomers are photoreduced ($FAD^{o-}$ $FAD^{o-}$) (Fig. 5j). This implies that the quantum yield for $FAD_{ox}$ to $FAD^{o-}$ photoreduction differs between the two L-Cry monomers. One monomer (referred to as "A" in Fig. 5j) acts as "very low intensity light sensor" with a high quantum yield $\Phi_A$. Hence, the very low photon number provided after 6 h of moonlight illumination is sufficient to photoreduce its flavin cofactor, resulting in the partially photoreduced $FAD^{o-}$ $FAD_{ox}$ moonlight state (Fig. 5j).

For direct comparison, our naturalistic moonlight's emission (in the main absorbance range of L-Cry: 330 nm–510 nm) is $5.4 \times 10^{10}$ photons/cm²/s (Supplementary Fig. 1e), which accumulates to

$1.2 \times 10^{15}$ photons/cm² in the 6 h required to reach the half-reduced moonlight state (Fig. 5d, e). For naturalistic sunlight, emitting $7.5 \times 10^{14}$ photons/cm²/s (330–510 nm), at least 5 min of sunlight illumination (i.e. $> 1.8 \times 10^{17}$ photons/cm²) are required to photoreduce the flavin in both L-Cry monomers in order to reach the fully photoreduced $FAD^{o-}$ $FAD^{o-}$ sunlight state (Fig. 5b, j). Thus, the second L-Cry monomer (monomer "B" in Fig. 5j) has a significantly lower quantum yield $\Phi_B$ for FAD photoreduction ($\Phi_B < \Phi_A$). Of note, even 12 h of naturalistic moonlight (~$2.3 \times 10^{15}$ photons/cm²) are about 100-fold below the minimal number of photons required to reach the fully photoreduced sunlight state, consistent with our observation that even after 12 h of naturalistic moonlight exposure only about half of the $FAD_{ox}$ molecules are photoreduced (Fig. 5e). Taken together, our results indicate the existance of L-Cry homodimers with very different light sensitivities of its two monomers, which enables the formation of a partially photoreduced moonlight state that is kinetically and structurally distinct from the fully photoreduced sunlight state of L-Cry (Fig. 5j, see Supplementary Fig. 6i and discussion for a more detailed model that also includes the possible formation of a short-lived [$FAD^{o-}$ $FAD_{ox}$] intermediate upon dark-/sunlight state interconversion and possible allosteric effects).

## Naturalistic sun- and moonlight differently affect L-Cry subcellular localization

In order to further investigate the response of L-Cry to naturalistic sun- and moonlight, we conducted Western blots and immunohistochemistry at different lunar and daily timepoints (Fig. 6a, a″). For the analyses of total protein levels via Western blots, we compared equal lengths of sun- versus moonlight illumination versus darkness, each having 8 h duration during their naturally occurring time (Fig. 6a, a″). L-Cry levels after 8 h of naturalistic sunlight (day before full moon = FM-1, diel time: zeitgeber time 8 = zt8, see Fig. 6a, a′) were significantly reduced compared to 8h under darkness at the same moon phase (FM-1, zt 0–10 min, Fig. 6b, c), in line with (canonical) L-Cry degradation in response to naturalistic sunlight.

In contrast to sunlight, exposure to an equal length (8 h) of naturalistic moonlight did not cause a reduction in L-Cry levels compared to an equivalent time (8 h) in darkness (FM-1, zt0–10 min versus FM7, zt0–10 min: Fig. 6b, c, Supplementary Fig. 7). Thus, any potential moonlight signaling via L-Cry occurs via a mechanism independent of L-Cry degradation.

We next examined the spatial distribution of L-Cry in worm heads (Fig. 6d) at lunar and diel timepoints (Fig. 6a–a″). After 8h of a dark night (i.e. NM, zt0–10 min), L-Cry is found predominantly in the nucleus of individual cells, (Fig. 6e–e‴, quantification as numerical data, i.e. nuclear/cytoplasmic ratio: Fig. 6h, for quantification as categorical data[38]: Supplementary Fig. 8a′–c″, d–f, Supplementary Data 1). Given that an equivalent time of 8h of sunlight exposure results in strong degradation of L-Cry and hence loss of staining signal (see Western blots above), we analyzed L-Cry's localization after a short exposure. After only 10 min of exposure to naturalistic sunlight (NM zt0 + 10 m, Fig. 6a, a′), the L-Cry nuclear localization strongly diminished, becoming predominantly cytoplasmic (Fig. 6f–f‴, numerical quantification Fig. 6h, categorical quantification Supplementary Fig. 8a′–c″, d–f,

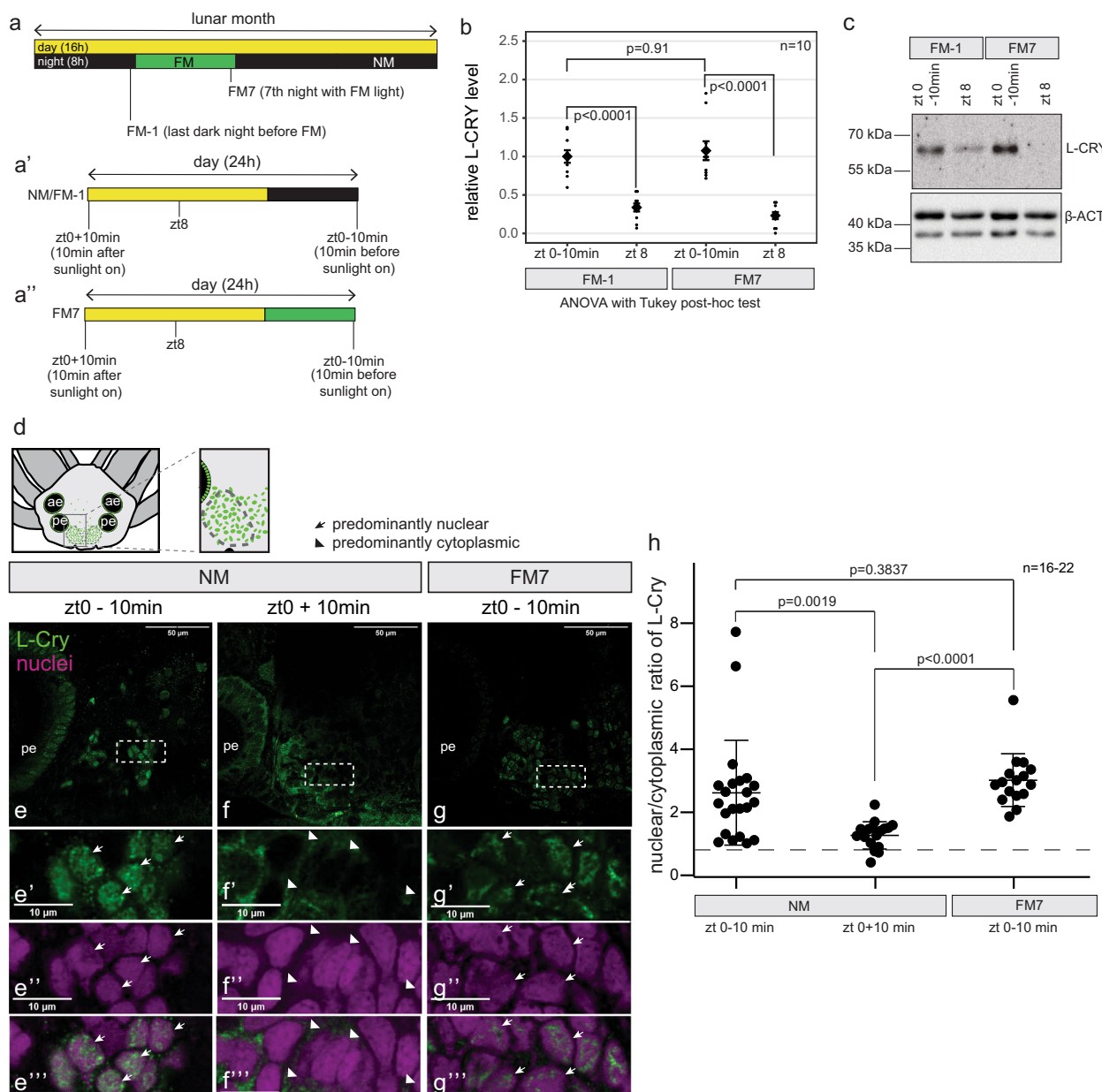

**Fig. 6 | Naturalistic moon- and sunlight impact differently on L-Cry localization and levels. a,a',a''** Overview of sampling timepoints. 16h day (light) and 8h night (dark or moonlight) per 24 h, with 8 nights of moonlight per month. NM and FM-1 have the same light regime, but are named differently for accuracy as they refer to different days relative to the lunar month. **b** relative L-Cry levels at indicated timepoints, as determined by western Blot. Individual data points as well as mean ± SEM are shown. Ordinary one-way ANOVA: $p < 0.0001$; adjusted $p$-values of Tukey's multiple comparison test: FM-1 zt0–10 min vs FM-1 zt 8: $p < 0.0001$, FM7 zt0–10 min vs FM7 zt8: $p < 0.0001$, FM-1 zt0–10 min vs FM7 zt 0–10 min: $p = 0.9141$. **c** Representative Western Blot used for quantification in (**b**), see Supplementary Fig. 7 for all other. In addition, in the context of further investigations such Western blots of identical or similar conditions were performed completely independently meanwhile more than three times with highly consistent results. **d** *P. dumerilii* head. Dashed ovals designate the oval-shaped posterior domains between the posterior eyes. Green dots: L-Cry+ cells. ae, anterior eye; pe, posterior eye. **e–g** Confocal single layer (1.28 μm) images of worm heads stained with anti-L-Cry antibody

(green) and HOECHST (magenta: nuclei). White rectangles: areas of the zoom-ins presented below. **e'–g'''** zoomed pictures of the areas depicted in e–g. Arrows: predominant nuclear L-Cry, arrowheads: predominant cytoplasmic L-Cry. Scale bars: 10 μm. Overview images with nuclear stain: Supplementary Fig. 8a–c. For a complete set of examples from a randomly chosen experimental repetition see Supplementary Fig. 10. In addition, in the context of further investigations such immunohistochemical stainings of worm heads from similar conditions were performed completely independently meanwhile more than three times with highly consistent results. **h** quantification of subcellular localization of L-Cry as nuclear/cytoplasmic ratio at indicated timepoints. Individual data points as well as mean ± SEM are shown. $p$ values: two-tailed $t$-test. NM zt0 –10 min vs NM zt0 + 10 min: $p = 0.0019$, NM zt0 –10 min vs FM7 zt0–10 min: $p = 0.3837$, NM zt0 + 10 min vs FM7 zt0 –10 min: $p < 0.0001$. Biological replicates: NM zt0 –10 min $n = 22$; NM zt0 + 10 min $n = 18$; FM7 zt0 –10 min $n = 16$. For quantification as categorical data, see Supplementary Fig. 8a'–f.

Supplementary Data 1, additional repetitions, image overviews and absolute cytoplasmic versus nuclear values: Supplementary Fig. 10). This suggests that naturalistic sunlight causes a shift of the protein to the cytoplasm, followed by degradation. (NM and FM-1 are identical in their illumination regime).

Given the degradation of L-Cry by naturalistic sunlight, we next asked the question if L-Cry is present at night timepoints, allowing for sufficient exposure to naturalistic moonlight to reach the moonlight state. We tested two diel timepoints of the first night lit by the naturalistic moonlight for circalunar entrainment (FM-1): at zt16 (just after the naturalistic sunlight is off and moonlight is on) and at zt20 (after 4 h of naturalistic moonlight exposure) (Supplementary Fig. 9a, a'). We observe that low levels of L-Cry can be detected at FM-1 zt16 (Supplementary Fig. 9b–b'''), and increase within the next hours (see FM-1 zt20, Supplementary Fig. 9c–c'''), with a predominantly nuclear L-Cry localization. At this timepoint still 4 h of moonlight illumination remain for the protein to biochemically reach the full moonlight state (ZT20 to ZT24). Based on these data we conclude that within the organism and under natural conditions (with the moon illuminating at least 8 h of the night under full moon conditions even during summer photoperiods), L-Cry has sufficient time to reach its moonlight state (by changing from sunlight to dark to moonlight state and/or by de novo synthesis of dark adapted L-Cry that reaches the moonlight state within 4 h- see biochemical kinetics, Fig. 5d–j, Supplementary Fig. 6f, g).

Upon further naturalistic moonlight exposure for seven continuous nights (FM7, zt0–10 min) L-Cry remained clearly nuclear (Fig. 6g–g''', numerical quantification Fig. 6h, categorical quantification: Supplementary Fig. 8f, Supplementary Data 1, additional repetitions, image overviews and absolute cytoplasmic versus nuclear values: Supplementary Fig. 10). Thus, the sunlight and moonlight-states of L-Cry correlate with distinct subcellular distribution patterns.

We occasionally observed that L-Cry at FM7, zt0–10 min can be more nuclear restricted than at zt0–10 min under NM (Supplementary Fig 8f, Supplementary Data 1). This is likewise apparent from the cytoplasmic values in the absolute quantifications (Supplementary Fig 10a). However, the large set of numerical quantification repetitions (Supplementary Fig. 10) also shows that this aspect can be more variable, possibly due to additional metabolic and/or sexual differentiation differences at present outside of our control. Thus, the exact cellular consequences between dark and moonlight states of L-Cry remain to be determined.

Complementing the spawning analyses on genetically mutated animals, these findings however clearly show that the two different light signals- moonlight versus sunlight- impact differentially on L-Cry quantity and localization.

This allows us to put forward a model, in which L-Cry directly via its biochemical states and connected cellular signaling properties is able to discriminate between (naturalistic) sun- and moonlight and to function as a gate keeper for potentially entraining light stimuli for the circalunar oscillator (Fig. 7b). But why would it be required to do this in nature? As we expand in more detail in the discussion, we speculate that this is necessary to entrain to a specific moon phase, which is the full moon phase for *Platynereis*. This moon phase is specifically characterized by the long duration of detectable moonlight, i.e. moonlight during the entire night[26] (Fig. 7a). Interestingly, this matches the biochemical kinetics of at least 6 h of light exposure to acquire L-Cry's biochemical moonlight state. However in nature, where the setting of the full and waning moons is immediately followed by sunrise (i.e. no darkness window, Fig. 7a[26],), measuring the duration of light exposure alone would not allow the worms to detect a specific moon phase (Fig. 7a). Thus, under the natural conditions of waning/waxing moon phases and sunrise/sunsets, being able to detect the switch from moonlight to sunlight is essential to determine the end of the moonlight phase and thus to discriminate between full moon and waning moon phases (Fig. 7a).

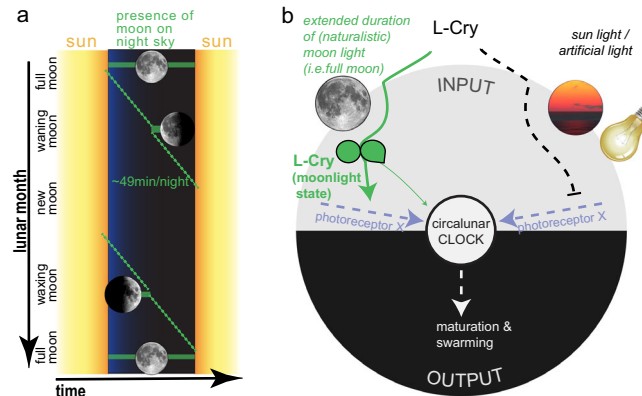

**Fig. 7 | The entrainment of the monthly oscillator requires the detection of a specific moon phase. a** Representation of the presence of the moon on the sky depending on moon phase. As a full cycle of the moon around the earth takes on average 24.8 h the presence of the moon relative to the sun shifts every night by ~49 min, indicated by the green diagonal. Worms need to specifically detect the full moon phase for circalunar oscillator entrainment, which requires that they can realize when a specific light (moonlight) starts and ends[26]. **b** L-Cry's function as moonlight duration and intensity detector for circalunar clock entrainment. L-Cry's biochemical property to only reach the full moonlight state after extended periods of (naturalistic) moonlight illumination allows for a discrimination of moonlight duration. As moon phases are characterized by the duration (and intensity) of the moon on the night sky, moonlight exposure duration translates into moon phase detection. When L-Cry is in its moonlight state it permits (thick green arrow) efficient entrainment of the circalunar oscillator via a yet unidentified photoreceptor X, while L-Cry (partly) blocks (dotted line) the light for circalunar entrainment when it is in its sunlight state. L-Cry might also provide minor information as entraining photoreceptor to the circalunar oscillator (thin green arrow).

Furthermore, L-Cry's gate keeping mechanism likely also makes the entrainment system more stable against irregular illumination as it could arise from thunderstorms.

## Discussion

Our work delivers the first molecular entry point into the mechanisms underlying a moonlight-entrained monthly oscillator. It uncovers L-Cry as a photoreceptor that can sense light ranging over five orders of intensity magnitude, due to the existence of (at least) two different stable states that can be both reached by light but with very different quantum efficiencies. Combined with different responses in the cell, this allows for a discrimination between sunlight and moonlight, as well as between different moon phases. The meaning of photons sensed during the day (i.e. sunlight) versus photons sensed during the night (i.e. moonlight) can be very different for the organism for their temporal ecological relevance. Thus, by measuring its intensity and duration, L-Cry can give light an ecological "valence", that is relevant to adequately adjust the worms' physiology and behavior (Fig. 7a, b). While we see the most apparent behavioral and physiological phenotypes of the *l-cry*−/− worms under artificial lab light conditions, these conditions are nevertheless highly informative about *l-cry's* role as ecological light valence detector. We interpret that this valence detection is under natural conditions necessary for the worms to synchronize to a specific moon phase: full moon. Full moon has the specific property that it is the moon phase during which the moon illuminates the entire night from sunset to sunrise (Fig. 7a). In order for the organism to identify this specific moon phase under natural conditions with waning and waxing moons as well as sunrise/sunsets (Fig. 7a), it needs to determine the duration not just of illumination, but specifically of the dim moonlight illumination. This discrimination is made by L-Cry. Under the artificial lab light conditions, the moonlight stimulus is much more intense and misses signals of the waning/

waxing moon phases. We hypothesize that under these artificial situations, the circalunar clock is still somewhat entrained, because there is no other entrainment stimulus it otherwise can entrain to. However, L-Cry signals that it is not really the "right" nocturnal light, which results in the observable, rather low population synchronization. When L-Cry is not present (as in the *l-cry−/−* worms), the nocturnal artificial light signal of the lab condition fully impacts on the circalunar clock. As it is (artificially) highly precise without possibly confusing waning and waxing moon signals, the entrainment results in the observed higher synchrony of the *l-cry−/−* population. If the nocturnal light signal mimics more closely the naturalistic full-moonlight, L-Cry permits its full impact on the circalunar oscillator, which results in the observed high population synchrony of wildtype worms under naturalistic lab light conditions. Furthermore, L-Cry's function as an ecological light valence detector also likely makes the entrainment system more stable against natural acute light disturbances, such as lightning.

At present, we can only speculate, how L-Cry can exert its valence function. We provide biochemical evidence that at dim light levels, corresponding to moonlight in nature, L-Cry can accumulate photons over time. L-Cry's photoreduction response to this accumulation is in its duration markedly different from its rapid, well-established response to strong light and non-linear, suggesting that a different moonlight signaling state might exist. Consistent with different L-Cry biochemical states under dim versus strong light, L-Cry under naturalistic moonlight is not following the conventional cytoplasmic degradation pathway, but accumulates in the nucleus (and diminishes from the cytoplasm). This suggests that different cellular compartments convey the different light messages to different downstream pathways.

Upon exposure to naturalistic daylight, L-Cry rapidly moves to the cytoplasm, where its protein levels become reduced, fully consistent with our previous data in S2 cells[15]. In *Drosophila melanogaster* dCRY is degraded via its light-induced interaction with the circadian clock protein Timeless and subsequent JETLAG (JET) ubiquitin-ligase–mediated proteolytic degradation[39]. Given that both Timeless and Jetlag exist in *Platynereis* (*Pdu*-Tim:[15], *Pdu*-Jet: reciprocal best blast hit in ESTs), it is possible that a similar mechanism exists in *Platynereis*. It is however noteworthy that *timeless* transcript levels are affected by room light in the bristle worm[15], indicative of additional levels of regulatory complexity.

In contrast to the suggested mechanism about L-Cry's fate in the cytoplasm based on existing knowledge from its *Drosophila* ortholog, the signaling downstream of nuclear L-Cry is at present completely enigmatic. It is clear from our data that neither darkness nor naturalistic moonlight causes degradation of nuclear L-Cry. This might indicate that ratios of cytoplasmic versus nuclear L-Cry could be important for circalunar clock moonlight entrainment. To complicate matters even further- it is possible that different spatial expression domains, such as eye versus brain, need to be considered separately in their responses to extended naturalistic moonlight and subsequent downstream signaling cascades.

In its role as a light-signal gatekeeper, only the accumulation of L-Cry molecules in the nuclear moonlight signaling state following prolonged moonlight exposure during full moon, would enable lunar entrainment via an additional photoreceptor "X", which by itself is not able to discriminate between the correct full moon signal and other "wrong" signals, such as sunlight or (in our lab experiment) the artificial/non-naturalistic nocturnal light source (Fig. 7b).

When L-Cry is photoreduced by light other than (naturalistic) moonlight, the light signaling of photoreceptor X towards the circalunar oscillator is inhibited (Fig. 7b). *L-cry* mutant worms lack this inhibitory mechanism, resulting in the observed (unnatural) synchronization to artificial moonlight. The somewhat better response of wildtype worms to naturalistic moonlight under re-entraining conditions indicates that the accumulation of moonlight state L-Cry not only

releases the inhibition but might enhance the activity of the yet to be identified photoreceptor X or provide additional light signaling by itself to the monthly oscillator.

Connected to the question of the transmission of the moonlight signal to the circalunar oscillator is also a better understanding of L-Cry's moonlight state. Is this state "just" a partial photoreduction of the state reached upon sunlight exposure or perhaps also a conformationally different state with distinct formation and decay kinetics? And what is the role of the L-Cry dimers? An intriguing observation is, that in presence of moonlight the moonlight state can be stably maintained over several hours, whereas the sunlight state completely reverts to the fully oxidized dark-state within minutes without accumulating the moonlight state while transitioning through partial photoreduction (Fig. 5j, Supplementary Fig. 6i). These different responses to moonlight illumination suggest that the moonlight- and sunlight states are conformationally and kinetically not equivalent.

We propose that partial FAD photoreduction in the moonlight state could be related to the formation of asymmetric L-Cry dimers, where one monomer retains oxidized FAD, while in the second monomer FAD is photoreduced to FAD$^{o-}$ (Fig. 5j; Supplementary Fig. 6i). This implies that the quantum yield for photoreduction of FAD$_{ox}$ to FAD$^{o-}$ differs between the two L-Cry monomers, i.e. different photon amounts are required to photoreduce the two flavins (Fig. 5j). Dim moonlight with its very low number of photons (-$10^{10}$ to $10^{11}$ photons/cm$^2$/s, Supplementary Fig. 1c, e) can photoreduce the first flavin when the protein was in its dark-state. This suggests that the first step has a high quantum yield, resulting in the build-up of the half-reduced moonlight state between 3 h and 6 h of illumination (mimicking full-moonlight at −4/−5m water depth, with 6 h yielding -$10^{15}$ photons/cm$^2$). Prolonged moonlight illumination even for 12 h is insufficient to photoreduce the second flavin of the L-Cry dimer, while (naturalistic) sunlight with its much higher photon numbers (>$10^{15}$ photons/cm2/s, Supplementary Fig. 1c, e) will result in full reduction within minutes. This suggests, that the transition from the state in which one flavin is already photoreduced to the state in which both flavins of the dimer are photoreduced, has a much lower quantum yield. Consequently, full photoreduction of dark-state L-Cry by sunlight would transition though a half-reduced state by first photoreducing the flavin with the higher quantum yield. Our spectra show, that naturalistic sunlight photoreduces about half of the flavins within about 30 s and moves towards full photoreduction after 110 s sunlight illumination (Fig. 5b). The number of photons provided by 30 s sunlight illumination (-$1.8 \times 10^{16}$ photons/cm$^2$, 330–510 nm) is more than sufficient for the first photoreduction, whereas at least -$6.5 \times 10^{16}$ to -$1.8 \times 10^{17}$ photons/cm$^2$ (110 s to 5 min sunlight exposure) are required to photoreduce the second flavin (Fig. 5b). Notably, the photon number provided by 12 h of naturalistic moonlight exposure (-$2.3 \times 10^{15}$ photons/cm$^2$, 330–510 nm) is well below the photon number required to start to photoreduce the second flavin of the L-Cry dimer.

Thus, the L-Cry dimer combines a highly sensitive "low light sensor" for moonlight detection with a much less sensitive "high light sensor" for sunlight detection. Notably, combinations of high light and low light sensors with different quantum efficiencies have been reported for other photoreceptors. For example, plant CRY1 is less light sensitive than its CRY2 homolog[40,41] and the LOV1 domain of plant phototropin1 is less sensitive than the LOV2 domain[42], which enables physiological responses of plants to bright and dim daylight. However, even though plant CRY2 members are frequently called "low light receptors", L-Cry's moonlight sensitivity exceeds their light sensitivity[40,43] by several orders of magnitude, while maintaining sensitivity to high-intensity sunlight.

In L-Cry, the formation of an asymmetric dimer containing two monomers with very different light sensitivities provides the molecular basis to sense and interpret light intensities across five orders of magnitude. Additionally, the L-Cry dimer enables the formation of a

third distinct partially photoreduced moonlight state in addition to the dark- and sunlight states. All these molecular properties of L-Cry are required to define the full moon phase as the moon phase with the highest moonlight intensity and longest duration on the night sky: The low light sensing L-Cry monomer detects moonlight (this study) and measures its intensity via the dose-dependent FAD photoreduction rate[36]. In turn, the moonlight duration is measured by the inter-conversion between the three L-Cry states. It starts with the formation of the moonlight state and either ends with the conversion of the moonlight state to the fully photoreduced sunlight state upon sunrise (full moon, waning moon) or with the conversion to the fully oxidized dark-state during waxing moon (Fig. 7a). (Note that when the moon rises in presence of or directly after sunlight, the sunlight state first decays to the dark-state before accumulating the moonlight state (Fig. 5h, i)). Finally, due to the slow build-up of the L-Cry moonlight state, the moonlight signaling state of L-Cry can only be formed during nights around full moon, when the moonlight intensity and duration are sufficient, which is essential for the synchronization of the worms to the full moon phase.

Regarding the nature of the three distinct L-Cry states, they most likely not only differ by the degree of FAD photoreduction observed by UV/VIS spectroscopy, but also by conformational and allosteric changes in the L-Cry protein (Supplementary Fig. 6i). Based on its sequence homology and the similarity of its FAD photoreaction to *Drosophila* CRY (dCRY), it is conceivable that L-Cry also displaces the regulatory C-terminal tail in the photoreduced state as observed for dCRY[37,44]. However, L-Cry homodimer formation may impact these conformational changes and lead to additional allosteric changes. These may further vary depending on whether moonlight or sunlight operates on the L-Cry homodimer (Supplementary Fig. 6i).

Furthermore, the molecular mechanisms underlying the fast decay (within min) of sunlight-state L-Cry to the fully oxidized dark state under moonlight conditions are unclear. Specifically, why does the half-reduced state not accumulate, as moonlight is able to maintain it? We propose that an allosteric regulation between the two L-Cry monomers may lead to the formation of a short-lived half-reduced [$FAD_{ox}$ $FAD^{\circ}$] intermediate state, that is conformationally and kinetically different from the "true" moonlight state and therefore does not accumulate in presence of moonlight (Supplementary Fig. 6i). Alternatively, sunlight photoactivation may induce structural changes in L-Cry, that prevent the formation of half-reduced $FAD_{ox}$ $FAD^{\circ-}$ dimer intermediates with a significant lifetime, and thereby the accumulation of the moonlight state (even in the presence of moonlight).

Certainly, more extensive mechanistic studies are required to further verify our models. However, our models are consistent with all our current in vitro data, and moreover, plausibly propose how the very different intensities of moon- and sunlight can lead to the formation of conformationally distinct dark-state (new moon), moonlight state (full moon) and sunlight state L-Cry proteins. Thereby L-Cry could translate different light qualities into different cellular signaling events, e.g. by changing L-Cry's subcellular localizations and cellular degradation rates (Fig. 6), to ultimately affect moonlight dependent physiology (Figs. 2–4).

Finally, an evolutionary consideration: Monthly synchronization by the moon has been documented for a wide range of organisms-including brown and green algae, corals, crustaceans, worms, but also vertebrates (reviewed in[6]). Furthermore, recent reports also provide increasing evidence that the lunar cycle influences human behavior (reviewed in[26,45]). Are the lunar effects mediated by conserved or different mechanisms?

When considering monthly oscillators with period lengths in the range of weeks, our implication of L-Cry as a light receptor in the circalunar entrainment pathway at first glance rather suggests that such monthly oscillator might not be conserved, given that direct L-Cry orthologs are not present in all the groups that are affected by

the lunar cycle[21]. However, taking further aspects into account, such a conclusion might be too premature. Could other members of the Cry/photolyase family take over similar functions? Furthermore, our entrainment data suggest the presence of additional moonlight entrainment photoreceptors, which might be conserved. Last, but not least the molecular mechanisms underlying the circalunar oscillator also await identification, and it is possible that conservation exists on this level. Examples are known from circadian biology and it will now require further work to reach a similar level of understanding for moon-controlled monthly rhythms and clocks.

## Methods

### Natural light measurements
Under water measurements of natural light at the habitat of *P. dumerilii* were acquired using a RAMSES-ACC-VIS hyperspectral radiometer (TriOS GmbH) for UV to IR spectral range. In coastal waters of the Island of Ischia, in the Gulf of Naples, the two radio-meters were placed on sand flat at 5 m depth near to *Posidonia oceanica* meadows, which are a natural habitat for *P. dumerilii*. Measurements were recorded automatically every 15 min across several weeks in the winter 2011/2012. To obtain a full moon spectrum, measurements taken from 10 p.m. to 1 a.m. on a clear full moon night on the 10 November 2011 were averaged. To subtract baseline noise from this measurement, a NM spectrum was obtained by averaging measurements between 7:15 p.m. to 5 a.m. on a NM night on 24 November 2011, and subtracted from the FM spectrum. Resulting spectrum: Supplementary Fig. 1a. To benchmark these moonlight spectra measured under water with moonlight measured on land, we compared the underwater spectra to a publicly available full moon spectrum measured on land on 14 April 2014 in the Netherlands (Supplementary Fig. 1a (red line), spectrum available at http://www.olino.org/blog/us/articles/2015/10/05/spectrum-of-moonlight). As expected, light with longer wavelengths was strongly reduced in the underwater measurements compared to the surface spectrum, since longer wavelengths penetrate water less efficiently. For the sunlight spectrum, measurements taken from 8 a.m. to 4 p.m. on a sunny day on 9 November 2011 were averaged.

### Naturalistic light systems (NELIS devices)
To emulate naturalistic sunlight and moonlight conditions, we employed NELIS (Natural environment light intensity system) (Marine breeding systems GmbH)[31]. The naturalistic moonlight device was composed of a combination of LEDs and an Ulbricht sphere for homogenous light mixing. For improved light distribution across the shelf, a naturalistic moonlight device was attached to each end of an acrylglass rod (two light sources and one rod per shelf). For details on the naturalistic sunlight source see ref. 31. Light spectra were measured using a ILT950 Spectroradiometer (International Light Technologies).

### Cloning and recombinant virus generation for L-Cry
Full length N-terminally His$_6$-tagged *P. dumerilii* L-Cry (1-567) was heterologously expressed in *Spodoptera frugiperda (Sf9)* insect cells using the Bac-to-Bac baculovirus expression system with the pCoofy27 expression vector. $1 \times 10^6$ *Sf9* cells were transfected with recombinant bacmid DNA using Cellfectin. The first generation P0 virus was harvested 3 days after bacmid transfection. A further virus amplification step was carried out and the P1 virus stock was used for protein expression. The volume of P1 virus stock to be added for sufficient protein expression was determined by test expression.

### Protein expression and purification
*Sf9* cells (#11496015, Thermo Fisher Scientific) were grown as suspension cultures in Sf-900 II media at 27 °C, 80 RPM. One litre of $1 \times 10^6$ *Sf9* cells/ml were transfected with P1 virus stock and incubated at 27 °C

for 72 h. Cells were harvested by centrifugation at 7000 rpm for 20 min and stored at −80 °C until purification. All purification steps were carried out in dark or dim red light conditions. Columns were wrapped with aluminum foil to avoid light-activation of L-Cry. The cell pellets were resuspended in lysis buffer (20 mM Tris pH 7.5, 150 mM NaCl, 20 mM imidazole, 5% glycerol, 5 mM β-mercaptoethanol) and lysed using a microfluidizer. The lysate was centrifuged at 27000 rpm for 45 min and the clarified supernatant incubated with nickel beads for 1 h. The nickel beads were loaded onto a batch column, washed with 50–100 mM imidazol and the L-Cry protein was eluted with 250 mM imidazole. Elution fractions containing L-Cry were concentrated, diluted with low salt buffer (50 mM Tris pH 7.5, 50 mM NaCl, 5% glycerol, 1 mM DTT) and loaded onto a 5 ml Hitrap Q sepharose anion exchange column (GE Healthcare). A gradient from 0–100 % high salt buffer (50 mM Tris pH 7.5, 1 M NaCl, 5% glycerol, 1 mM DTT) was applied. L-Cry containing fractions were pooled, concentrated and loaded onto a HiLoad S200 16/60 size exclusion chromatography (SEC) column (buffer 25 mM Bis-tris propane pH 8.0, 150 mM NaCl, 5% glycerol, 1 mM TCEP). Fractions containing pure L-Cry were pooled, concentrated to 10 mg/ml and snap frozen in liquid nitrogen for storage at −80 °C. 2 mg of L-Cry was obtained from 10 g of pellet. The identity of the L-Cry protein was confirmed by mass spectrometry.

### Reverse-phase HPLC analyses of the chromophore content of L-Cry

Flavin mononucleotide (FMN), flavin adenine dinucleotide (FAD) and methenyltetrahydrofolate (MTHF) were dissolved in buffer (25 mM Bis-Tris propane pH 8.0, 150 mM NaCl, 5% glycerol) and run at 1 ml/min (20 °C) over a Macherey–Nagel C18 Gravity-SB (150/4/5 μm) column to separate the chromophores by reverse phase (RP) HPLC analyses. A gradient from 20–100% of methanol against water (+0.1% trifluoroacetic acid) was used for optimal separation. To analyze the chromophore content of L-Cry, purified L-Cry was heat-denatured for 5 min at 97 °C and centrifuged at 14,000 RPM for 10 min at 4 °C. The supernatant was subjected to RP-HPLC analysis. The chromophores were monitored by absorption at 370 nm.

### Analytical size exclusion chromatography (SEC) and SEC coupled with multi-angle light scattering (SEC-MALS)

Analytical SEC of dark-state L-Cry was carried out on a S200 10/300 size exclusion column (SEC buffer 25 mM Bis-Tris propane pH 8.0, 150 mM NaCl) under red light conditions. SEC-MALS was carried out to determine the exact molecular weight and oligomeric state of purified L-Cry based on the SEC elution volume and light scattering. For SEC-MALS, purified L-Cry was loaded onto a Superose 6 10/300 size exclusion column and run at a flowrate of 0.4 ml/min in SEC buffer. MALS data were obtained from a DAWN DSP instrument (Wyatt Tech, Germany) and processed using ASTRA 4.90.07. Elution volumes and corresponding molecular weight of calibration standards: 10.3 ml to 670 kDa (thyroglobulin), 13.67 ml to 158 kDa (γ-globulin), 15.71 ml to 44 kDa (ovalbumin), 17.42 ml to 17 kDa (myoglobin) and 20.11 ml to 1350 Da (vitamin B12).

### UV/VIS spectroscopy on L-Cry: blue light-, sunlight- and moonlight photoreduction and dark recovery

UV/Visible absorption spectra of the purified L-Cry protein in final SEC purification buffer (25 mM Bis-tris propane pH 8.0, 150 mM NaCl, 5% glycerol, 1 mM TCEP) were recorded on a Tecan Spark 20 M plate reader unless otherwise stated. A light-state spectrum of L-Cry with fully photoreduced FAD⁰⁻ was collected after illuminating dark-adapted L-Cry for 110 s using a four-chip blue-light (450 nm peak) LED with a blue-light dominated spectrum (PerkinElmer ACULED Dyo: datasheet- Supp. Material 1; for our own measurements see Supplementary Fig. 1d, d′,e; $6.21 \times 10^{16}$ photons/cm²/sec). To analyze sunlight- and moonlight-dependent FAD photoreduction, dark-adapted L-Cry

(kept on ice) was continuously illuminated with naturalistic sunlight (Supplementary Fig. 1c; $1.55 \times 10^{15}$ photons/cm²/sec at the sample) or naturalistic moonlight (Supplementary Fig. 1c–e; $9.65 \times 10^{10}$ photons/cm²/s at the sample) and UV–VIS spectra (300–700 nm) were collected at different time points.

Dark recovery kinetics (FAD reoxidation) of L-Cry at 18 °C following illumination with light with a blue-light dominated spectrum (110 s), sunlight (20 min on ice) or moonlight (6 h on ice) were measured by recording absorbance changes at 450 nm over time or by extracting 450 nm absorbance values from complete UV/VIS spectra. To measure dark recovery kinetics on ice, complete UV–VIS spectra (300–700 nm) were collected at different time points following 110 s light with a blue-light dominated spectrum - or 20 min sunlight illumination and absorbance values at 450 nm were extracted from the full spectra (sample was kept on ice and in darkness between measurements). Additionally, a temperature-controlled Jasco V-550 UV–VIS spectrophotometer was used to determine dark recovery kinetics of L-Cry (after 110 s light with a blue-light dominated spectrum) at 6 °C based on absorbance changes at 450 nm. The time constants for dark recovery were calculated by fitting a single exponential curve to the experimental data. To assess if sunlight can further increase FAD photoreduction starting from the moonlight activated state, L-CRY was first illuminated with continuous moonlight for 6 h, followed by 20 min of sunlight illumination (on ice). Complete UV–VIS spectra from 300–700 nm were measured in each case. Spectra were analyzed using Origin (Version 7.5/10.5.(trial); OriginLab Corporation, Northampton, MA, USA). When baseline shifts occured between individual spectra, e.g. due to condensation and evaporation upon long-term measurements and cooling on ice between measurements, absorbance values were normalized at a specific wavelength (Y-axis labeled "Normalized absorbance"). Otherwise absorbance values are shown as directly measured (Y-axis labeled "Absorbance").

### Recovery of L-Cry dark-state in presence of moonlight

To assess if moonlight can maintain the light state, L-Cry was initially illuminated with sunlight for 20 min or with light with a blue-light dominated spectrum for 110 s, followed by continuous moonlight illumination up to 6 h with the sample kept on ice. Complete UV–VIS spectra (300–700 nm) were collected at different time points. Absorbance values at 450 nm were taken from the complete spectra obtained between 5 min and 2 h 30 min moonlight exposure and used to determine the time constant for recovery of oxidized FAD after light with a blue-light dominated spectrum - or sunlight induced photoreduction in presence of moonlight.

### Sunlight illumination of moonlight activated L-Cry

To assess if sunlight can further increase FAD photoreduction starting from the moonlight-activated state, L-CRY was first illuminated with continuous moonlight for 6 h, followed by 20 min of sunlight illumination (on ice). Complete UV–VIS spectra from 300–700 nm were measured in each case.

### Worm culture

*P. dumerilii* were grown as previously described[15,46]. All animal work was conducted according to Austrian and European guidelines for animal research. As these are invertebrates without special features grown in the lab for generations there are no ethical approvals required. Photoperiod 16:8 (L/D), circalunar entrainment: nocturnal light for 6–8 nights (see figure legends for each experiment) every 29 to 30 days (centering around full moon ("inphase") or new moon ("outphase") in Vienna. Light spectra and intensities of Supplementary Fig. 1b, c were measured with a recently calibrated ILT950 spectrometer (International Light Technologies Inc Peabody, USA) and converted to photons/cm²/s.

## Generation and genotyping of *l-cry* KO worms

Design and construction of TALENs targeting *l-cry* is described in ref. 28. For genotyping of TALEN mediated mutagenesis, genomic DNA (gDNA) extraction of immature and premature, anaesthetized (7.5% MgCl₂-/H₂0, 1:1 diluted with sea water) worms was conducted by cutting 5–10 trunk segments with a scalpel and incubating the trunk segments in 45 µl 50 mM NaOH at 95 °C for 20 min. After adding 5 µl of 1 M Tris/HCl pH 7.5, cell debris was separated by centrifugation and the clear gDNA containing supernatant was used as template for the PCR reaction.

To isolate gDNA from mature animals, worms were frozen at −20 °C and gDNA later extracted using "NucleoSpin Tissue Mini kit for DNA from cells and tissue" (Macherey–Nagel) exactly as recommended by the manufacturer.

PCR was performed with OneTaq Quick-Load 2x Master Mix with Standard Buffer (New England Biolabs). PCR products were visualized on an agarose gel and the genotype was determined on size (168 bp: wildtype, 134 bp: Δ34 + Δ9 mutant allele, 157 bp: Δ11 mutant allele).

| Primer | Sequence 5'–3' |
|---|---|
| l-cry_fwd | AAGAGAAGACTGACGATTGGGAC |
| l-cry_rev | CTGCAACTTCCCCATCCC |

Primers used for l-cry genotyping.
Full length *l-cry* cDNA- GenBank: MW161054.

## Monoclonal antibody production

A peptide consisting of amino acids 52–290 of L-Cry protein (GenBank ID: MT656570, predicted size 25 kDa) was cloned and expressed in bacteria cells. Subsequently, this peptide was purified and used for mouse immunization, thereby acting as epitope in production of a monoclonal antibody against L-Cry. Upon screening of multiple clones, two clones (4D4-3E12-E7 and 5E3-3E6-E8) were selected and used in combination. Monoclonal antibodies were produced by and purchased from the Monoclonal Antibody Facility at Max Perutz Labs (Medical University of Vienna, Department of Medical Biochemistry).

## Immunohistochemistry, microscopy and L-Cry localization determination

Worm heads were dissected with jaws and fixed in 4% PFA for 24 h at 4 °C. Samples were subsequently permeabilized using methanol, digested for 5 min with Proteinase K at room temperature without shaking and post-fixed with 4% PFA for 20 min at room temperature. Next, samples were washed 5 times for 5 min with 1x PTW (1xPBS/ 0,1% TWEEN20) and incubated in hybridization mixture[47] used in in situ hybridization protocol, at 65 °C overnight. Worm heads were washed with 50% formamide/2X SSCT - standard saline citrate containing 0.1% Tween20 (Sigma–Aldrich) (2×, 20 m), then with 2X SSCT (2×, 10 m) and with 0.2X SSCT (2×, 20 min); all washing steps at 65 °C. After blocking for 90 min with 5% sheep serum (Sigma–Aldrich) at room temperature, samples were incubated in L-Cry antibodies 5E3-3E6-E8 (1:100) and 4D4-3E12-E7 (1:50) in 5% sheep serum (Sigma-Aldrich). Secondary antibody, Cy3 goat anti-mouse IgG (A10521, Thermo Fisher Scientific) was diluted 1:400 in 2.5% sheep serum (Sigma-Aldrich). Incubations were done for at least 36 h at 4 °C shaking and after each incubation time, samples were washed with 1× PTW three times for 15 min at room temperature and a fourth time overnight at 4 °C. After this, Höchst 33342 (H3570, Thermo Fisher Scientific), diluted 1:2000, was added for at least 30 min at room temperature. Samples were then washed three times for 15 min with 1× PTW at room temperature and mounted with 25 mg/ml DABCO (Roth/Lactan) in an 87% glycerol (Sigma–Aldrich) solution. All solutions were made with 1× PTW (PBS + 0.1% Tween 20). Heads were imaged on a Zeiss LSM 700 laser scanning confocal microscope using LD LCI Plan-Apochromat 25X, Plan-Apochromat 40X

by CHD: T-PMT detection system and Zeiss ZEN 2012 software. Lasers: DAPI 405 nm and Cy3 555 nm.

Categorical scoring: Using Fiji/ImageJ[48], nuclear outlines were marked as Regions Of Interest (ROI) on the 405 nm channel images (Höchst staining). ROIs were then used for scoring of the signal localization (inside = nucleus versus outside = cytoplasm) on the 555 nm channel of the same images (L-Cry).

Quantitative scoring: Using the deep learning-based image segmentation algorithm Cellpose[49] on the 405 nm channel images, the Hoechst-stained nuclei were identified and marked as Regions Of Interest (ROI). L-Cry signal was then determined for these nuclear ROIs using in Fiji/ImageJ[48]. Signal intensity was determined by calculating Corrected Total Cell Fluorescence (CTCF) using the formula CTCF = Area (ROI_1)*Mean (ROI_1)-Area (ROI_1)*Mean(ROI_(background ROIs)). A sum of CTCF values of all the nuclei was subtracted from the CTCF value of the whole brain area, to obtain the corresponding value for non-nuclear, i.e. cytoplasmic signal. Finally, the ratio between nuclear and non-nuclear (cytoplasmic) signal intensity was calculated for corresponding regions of different worm heads to compare between different ZTs.

## Protein extraction and western blots

Per biological replicate four premature worms were anaesthetized (7.5% MgCl₂-/H₂0, 1:1 diluted with sea water), decapitated and heads transferred to a 1.5 ml tube containing 200 µl RIPA lysis buffer (R0278 Sigma–Aldrich), 10% Triton X100 and protease inhibitor mix (cOmplete Tablets, EDTA-free, *EASYpack*, Roche). The tissue was homogenized by grinding with a cone-shaped pestle, all steps on ice. Cell debris was pelleted by centrifugation 4 °C. Protein concentration of lysates was determined using Bradford reagent (BIORAD). Protein extract were separated by SDS-PAGE (10% acrylamide) and transferred (transfer buffer: 39 mM glycine, 48 mM Tris, 0.04% SDS, 20% MetOH) to a nitrocellulose membrane (Amersham™ Protran™ 0,45 µm NC, GE Healthcare Lifescience). Quality of transfer was controlled by Ponceau-S (Sigma–Aldrich) staining of the membrane. After 1 h of blocking with 5% skim milk powder (Fixmilch Instant, Maresi) dissolved in 1× PTW (1× PBS/0.1% Tween20) at room temperature, the membrane was incubated with the appropriate primary antibody diluted in 2.5% milk/PTW at 4 °C overnight. [anti-L-Cry 5E3-3E6-E8 (1:100) and anti-L-Cry 4D4-3E12-E7 (1:100); anti-beta-Actin (Sigma, A-2066, 1: 20,000)]. After 3 washes with 1× PTW the membrane was incubated with the species-specific secondary antibody [anti-Mouse IgG-Peroxidase antibody, (Sigma, A4416, 1:7500); Anti-rabbit IgG-HRP-linked antibody (Cell Signaling Technology, #7074, 1:7.500] diluted in 1× PTW/1% skim milk powder for 1 h, RT. After washing, SuperSignal™ West Femto Maximum Sensitivity Substrate kit (Thermo Fisher Scientific) was used for HRP-signal detection and finally signals were visualized by ChemiDoc Imaging System (BIORAD). Specific protein bands were quantified in "Image J" and L-Cry protein levels were normalized to beta-Actin.

## Collection and analysis of spawning data

Worm boxes were checked daily for mature worms. Worms which had metamorphosed into their sexually mature male or female form and had left their tube to perform their nuptial dance were scored as mature animals.

The recordings of mature animals in nature (collected from June 1929–1930 in Naples[34,50]) were digitalized and all months were aligned to relative to the same moon phase and combined. For comparisons of these data with our spawning data from the lab, we aligned the first day after full moon in nature with the last day of full moon stimulus in the lab, since *P. dumerilii* synchronizes its circalunar clock to the end of the full moon stimulus[5].

For analysis, each day of the lunar month was assigned a number from 1 to 30. For linear plots, the percentage of mature worms per

lunar day was then plotted as a histogram. The spawning distributions of two conditions were compared using the Kolmogorov–Smirnov Test. For the circular analysis[51–53] of spawning data, the lunar day of spawning was multiplied by 12 for each worm, so that the 30 lunar days regularly distributed on the 360° circle. Each dot represents one mature worm unless stated otherwise. Circular data can be described using the mean vector (displayed as an arrow), which is defined by its direction angle (μ) and its length (r). The direction angle μ is given relative to 0° (moon off). The value of length r (also called phase coherence) ranges from 0 to 1, where higher values indicate higher phase coherence (i.e synchrony). In order to test, if the observed data distribution is significantly different from random, we performed the Rayleigh's Uniformity Test and used $p < 0.05$ as cutoff for significance. Non-uniform distribution is consistent with lunar rhythmicity. For comparing two circular datasets (e.g. of different genotypes or different months in the phase shift experiments), we used the non-parametric Mardia–Watson–Wheeler test. Circular analysis of these data was performed using Oriana (Version 4.02, Kovach Computing Services).

### Phase-shift experiments

For Phase-Shift experiments, boxes with adult worms (at least 3 months old) were transferred from standard light conditions (see "worm culture") to the naturalistic light systems (sun- and moonlight) mounted in light-tight black shelves. Number of mature worms was recorded daily and number of mature worms per day was used to calculate percentage of mature worms (one month: 100%). Data were smoothened using a rolling mean with a window size of 3 days. Data analysis was performed as described in "Collection and analysis of spawning data".

### Whole mount in situ hybridization combined with immunohistochemistry for L-Cry

Probes were generated de novo using previously cloned plasmids as template. Genes of interest were amplified via PCR using Phusion Polymerase (NEB) and primers for pJET 1.2 with an overhang of Sp6 promoter. PCR product was purified following the protocol for "QIAquick PCR Purification Kit" (Qiagen). Agarose gel electrophoresis showed right amplicon sizes and single bands. For the in vitro transcription, 1 μg of linearized template was used. Riboprobes were labeled with Digoxigenin-coupled to UTP (Roche Diagnostics) and transcribed with Sp6 polymerase at 37 °C for 4 h. Probes were purified according to the "RNeasy Kit" (QIAGEN) and eluted in 40 μl RNase-free water. 600–1000 ng of the riboprobes were used.

Whole mount in situ Hybridization was carried out on premature worms of the RE strain, following published procedures[47,54] with adjustments made to combine it with L-Cry immunohistochemistry. Worm heads were fixed in 4% PFA for 2 h at RT while shaking. Proteinase K digest: 5 min, during which samples were very slightly rocked. After blocking in 5% sheep serum/1X PTW, worm heads were incubated with the monoclonal L-Cry antibodies (5E3-3E6-E8 diluted 1:100 and 4D4-3E12-E7 diluted 1:50), anti-Digoxigenin-AP coupled antibody Fab fragments (Roche Diagnostics, 11093274910) and sheep serum diluted to 2.5% with 1× PTW for 36–40 h at 4 °C, shaking. After detection using NBT/BCIP, samples were incubated in the secondary antibody Alexa Fluor-488 goat anti-mouse IgG (Thermo Fisher Scientific), 1:400 (36–40 h at 4 °C, shaking), washed in 1× PTW and mounted in DABCO/Glycerol. Imaging was done using Axioplan Z2 Microscope (Carl Zeiss) with AxioCam MRc5 color CCD camera (Carl Zeiss) and captured using ZenPro Software (Carl Zeiss). The images were edited with either ImageJ or Photoshop CC.

### Statistical analysis

All data analysis was conducted using R 3.6.1[55], GraphPad Prism 8.4.2, Oriana 4.02 and Microsoft Excel 2010.

### Reporting summary

Further information on research design is available in the Nature Research Reporting Summary linked to this article.

## Data availability

Source data are provided with this paper in the Suppl. Figures, Supplementary Data 1 and the Source Data file (includes a readme file for overview). Full length *l-cry* cDNA- GenBank: MW161054, L-Cry fragment sequence for antibody production: GenBank ID: MT656570. The full source data can also be downloaded from DRYAD/Zenodo: https://doi.org/10.5061/dryad.wm37pvmq8 and https://doi.org/10.5281/zenodo.6779947 Source data are provided with this paper.

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

## Acknowledgements

We thank the members of the Tessmar-Raible and Wolf groups for discussions. Andrej Belokurov, Margaryta Borysova and Netsaneh Getachew for excellent worm care at the MFPL aquatic facility. We are grateful for support by the IMB Media Lab, Protein Production and Proteomics Core Facilities. We would like to thank Dr. Svenja Morsbach and Beate Müller from the Max Planck Institute for Polymer Research for helping with HPLC experiment. This study was supported by the following funding sources: the European Research Council under the European Community's Seventh Framework Programme (FP7/2007–2013) ERC Grant Agreement 337011 and the Horizon 2020 Programme ERC Grant Agreement 819952 and the HFSP research grant (#RGY0082/2010) to K.T-R, the research platform 'Rhythms of Life' of the University of Vienna and the Austrian Science Fund (FWF, http://www.fwf.ac.at/en/): SFB F78 to K.T.-R. the FWF (project nr: I2972), and the European Research Council (ERC, project #260564) to F.R., a Lise-Meitner fellowship by the Austrian Science Fund (FWF, project nr: M2820), a DFG fellowship of the Excellence Initiative by the Graduate School Materials Science in Mainz (GSC 266) to S.K. M.Z. and B.P. were enrolled in the Vienna Doctoral School in Cognition, Behavior, and Neuroscience of the University of Vienna. None of the funding bodies was involved in the design of the study, the collection, analysis, and interpretation of data or in writing the manuscript.

## Author contributions

B.P.,S.K. Conceptualization; resources; data curation; formal analysis; validation; investigation; visualization; methodology; writing—original draft; review & editing. M.Z. Conceptualization; resources; data curation; formal analysis; validation; investigation; visualization; methodology; A.C., D.R., N.S.H, E.A., E.J., L.O. resources; data curation; formal analysis; validation; investigation; visualization; methodology. F.R. Conceptualization; supervision; funding acquisition; validation; investigation; visualization, writing—original draft, review & editing. E.W. and K.T-R Conceptualization; resources; data curation; formal analysis; supervision; funding acquisition; validation; investigation; visualization; methodology; writing—original draft; project administration; writing—review & editing.

## Funding

## Competing interests

The authors declare no competing interests.
