## [Peer Review File · Nature Communications]

Title: A Cryptochrome adopts distinct moon- and sunlight states and functions as sun- versus moonlight interpreter in monthly oscillator entrainmentREVIEWER COMMENTS

Reviewer #1 (Remarks to the Author):

Firstly, apologies to the authors for taking my time to review this manuscript. However, I enjoyed doing so and it is my view that the paper provides a compelling and rather elegant argument for the role of light receptive cryptochrome (L-Cry) in discriminating between solar and lunar light signals and lunar spawning entrainment in *Platynereis*.

For a while now it has been known that lunar as well as daily (circadian) clock mechanisms exist together in some marine species, one of which is the polychaete, *Platynereis dumerilii* studied here. Any mechanistic basis of circalunar timing has yet been missing from our understanding, although lunar rhythms are known now to be critically important for marine organisms, but also terrestrial ones too. Here the authors tapped into some really cute observations on differences in the efficiency of artificial moonlight entrainment between lab-entrained worms and naturally entrained animals. The authors explore the hypothesis that light receptive cryptochrome plays a role in circalunar rhythm entrainment and created mutant worms using TALENs to induce frameshift and premature stop codons in exon 3 of *l-cry* as a foundation to this exploration. These mutations were very rigorously validated by specific immunochemical and molecular genotyping approaches, aided by the production of monoclonal L-Cry antibodies. The authors performed some very well executed behavioural studies to compare entrainment (spawning/maturity of worms was the observed phenotype) on mutant and wt animals under artificial sun and moonlight condition. The surprising improvement in entrainment of worms lacking L-Cry under standard lab conditions sparked imaginative and well executed lunar phase shift experiments demonstrating that natural moonlight conditions (rather than lab moonlight which is rather more intense) improved phase-shifting/spawning synchrony in wildtype animals, more inline with that exhibited by *l-cry*^{-/-} animals under lab (harsh lighting) moonlight but less so in mutant animals. As such, and considering these observations, the authors proposed that L-CRY acts as 'gatekeeper' of light- 'blocking' the 'wrong' light from influencing the lunar clock and has a rather minor role as a photoreceptor.

Then using biochemical *in vitro* approaches it was shown that L-CRY discriminates between sun and moonlight by forming photo-reduced states differently (in terms of duration and intensity of light required to induce reduction)- i.e. moonlight and sunlight states. These experiments seemed very well done and well controlled and may turn out to be really important in our fundamental understanding of how organisms can distinguish sunlight from moonlight to drive solar and lunar rhythms. A really stunning finding was that, *in vitro*, moonlight takes about the same time to induce photoreduction of the FAB chromophore (about 6h) as it does for *in vivo* lunar entrainment!

Finally, the authors used immunochemical (immunohistochemistry and Western blotting) strategies to show light degradation dynamics of L-Cry and, intriguingly the differential nuclear and cytoplasmic localisation of L-Cry under moonlight (nuclear) and sunlight (cytoplasmic) conditions. Again, these data look convincing and well reported.

Together the data have enabled the authors to propose a tantalising and testable model whereby the intensity and duration differences of sunlight and moonlight (natural) are registered differently by L-Cry (biochemically and in its sub-cellular localisation) to influence lunar entrainment in this worm.

These findings are exciting; they are some of the very first to show how light signals may entrain a non-circadian phenotype and open a door into the circalunar clock- The model proposed is testable and needs following up. Whilst some of the discussion is speculative (and acknowledged as such) the paper is a certainly a significant advance in our understanding of the mechanistic basis of rhythmicity that is crucial to a hugely diverse organisms in the marine and terrestrial realms. As such the findings go beyond the chosen model worm and may have much wider impacts on our understanding of other circalunar organisms across a wide range of different taxa.

For the reasons given above, I think the manuscript should be published and have only minor (some very minor) queries/comments that might be considered:

- 1) Lines 195-197 state that according to the Mardia-Watson-Wheeler test there is less or NO significance in mutant worms before and after entrainment. I can't really see this in the figure/table. Perhaps the authors could elaborate or indicate in the main text which values indicate this?
- 2) Line 202 'stronger'this doesn't read too well- perhaps change to something like ' have shifted more robustly in response to.....or something like?
- 3) Line 220 ...'function itself IS a sensor...'
- 4) Figure 4d to 4d''' (not referred to in text). Comparing 4d to 4d''' (month 1 to month D) Unless, I am mistaken, month D has no significant deviation from uniformity ($p=0.07$). Can the authors say whether it is OK to consider this as re-entrainment in their analyses/interpretation if the shifted animals are not spawning at a particular phase of the lunar month?
- 5) Line 394- Delete 'Already'. Or, say, "After only 10 minutes...."
- 6) Line 404- delete 'already'.
- 7) Line 419- delete 'also'.
- 8) Line 552- I suggest changing 'quick' to 'too hasty' or 'premature'.
- 9) Lines 673-679. I didn't find this section very easy to follow. Was the supernatant column purified before PCR? Please clarify.
- 10) Line 694- define PTW here, not later.
- 11) Line 732- 'skim' or 'skimmed' milk??
- 12) Line 777- anti-DIG or just DIG-UTP?
- 13) The references have some formatting errors- such as species names lacking necessary formatting (italics etc). Please check.

David Wilcockson

Reviewer #2 (Remarks to the Author):

The manuscript presents a study on the moonlight entrainment of the worm *Platynereis dumerilii*. This moonlight aspect of light entrainment is of high interest to the clock community, in particular with the molecular focus presented here. The photoreceptor cryptochrome is identified to govern the lunar clock and to help a differentiation between sun light and moon light levels. The experiments have been carefully conducted and demonstrate the importance of applying correct light sources with respect to both intensity and spectral distribution. The data are strong and support an assignment to

cryptochrome. A further strength of the study is the evaluation of the effects of light on many different levels, from molecular data on purified protein to localization studies to behavioral assays on living knockout mutants. Serious concerns remain with respect to the interpretation, which is biophysically invalid. I additionally object to the novelty of 'the new concept that a light receptor does not just sense light, but by its intensity and duration can give light a valence'.

1. Abstract. The term 'valence interpreter' is nowhere explained in the manuscript. It is unnecessary to invent a new term for something available in the literature as 'high light' and 'low light' reception. This discrimination is commonly found in plant photoreception, for example for phototropin. Moreover, our eyes perform the same task by using rods and cones with different sensitivity to the same wavelength of light.

2. Line 72. The statement concerning previous work on the L-Cry is exaggerated. Reference 15 does not support 'that Pdu-L-Cry functions in principle as a light receptor in a heterologous cell expression system.' Please rephrase more carefully. Most evidence in ref. 15 is shown on tr-Cry and only very minor experiments have been performed on L-Cry. Moreover, the presence of tr-Cry should be included in the manuscript together with a differentiation in roles.

3. Figure 5j. The scheme requires fundamental revisions. Sun light will also populate the moonlight state, only that the intensity is so high that it is further converted. The same applies to the recovery. Under moonlight conditions, the sunlight state will recover to the dark state via the moonlight state, only that the recovery from the moonlight state to the dark state is much faster and therefore the moonlight state does not accumulate.

4. L. 335. The idea is very reasonable to correlate the half-reduced state in the spectra with a homodimer with only one out of two flavins being converted. However, the difference between the two homomers is not the energy difference of the transition but the quantum yield. Sun light has the same energy as moon light when absorbed by FAD at 450 nm! (Revise extended data Fig. 6i accordingly) Only the number of photons is different. Therefore, one homomer has a high quantum yield, corresponding to a highly sensitive receptor. The second homomer has a low quantum yield, accordingly, many photons are required to convert the FAD to the anion radical. Differences in quantum yield of flavin are very common for LOV domains in phototropin or aureochrome and have been documented as well for bird and plant cryptochromes. Please do not involve redox potentials (l. 529), these are certainly a different issue.

5. L. 325. The direct comparison of light levels in nature with those applied to a purified protein requires to take scattering and absorption by the tissue of the worm into account. How have these contributions been included in the setting of the light level?

6. Figure 5a, line 289. The light scattering data (MALS) need to be converted into corresponding molecular weights. The panel does not reveal the experimental results. An example for how to analyze SEC-MALS results is shown in Winkler et al. (2013) Nat. Struct. Mol. Biol. 859 Figure 1b.

7. L. 354. The fact that the moonlight state is not detected in the recovery does not allow for the conclusion that it is not formed in the recovery. It is simply not strongly populated which might be attributed to an allosteric regulation of the two homomers.

Minor points.

- Chemical compounds should be lower case not upper case throughout the manuscript.
- Replace 'adduct' with 'educt' throughout the manuscript.
- If all in vitro experiments were performed in the presence of the strong reductant TCEP, this fact needs to be included in the main text.
- Extended data Figure 1d. The emission of the LED at 600 nm is zero not 10^{12} photons / (cm² s). A measurement needs to be included with the LED switched off to demonstrate the limit in sensitivity of the detector and to generate a zero line.
- Figures 5f and 5g. The right axis is corrupted and not at the same level as the left axis.

Point-to-point responses to reviewers' comments

Our responses are written in italics and purple color.

REVIEWER COMMENTS

Reviewer #1 (Remarks to the Author):

Firstly, apologies o the authors for taking my time to review this manuscript. However, I enjoyed doing so and it is my view that the paper provides a compelling and rather elegant argument for the role of light receptive cryptochrome (L-Cry) in discriminating between solar and lunar light signals and lunar spawning entrainment in *Platynereis*.

For a while now it has been known that lunar as well as daily (circadian) clock mechanisms exist together in some marine species, one of which is the polychaete, *Platynereis dumerilii* studied here. Any mechanistic basis of circalunar timing has yet been missing from our understanding, although lunar rhythms are known now to be critically important for marine organisms, but also terrestrial ones too. Here the authors tapped into some really cute observations on differences in the efficiency of artificial moonlight entrainment between lab-entrained worms and naturally entrained animals. The authors explore the hypothesis that light receptive cryptochrome plays a role in circalunar rhythm entrainment and created mutant worms using TALENs to induce frameshift and premature stop codons in exon 3 of *l-cry* as a foundation to this exploration. These mutations were very rigorously validated by specific immunochemical and molecular genotyping approaches, aided by the production of monoclonal L-Cry antibodies. The authors performed some very well executed behavioural studies to compare entrainment (spawning/maturity of worms was the observed phenotype) on mutant and wt animals under artificial sun and moonlight condition. The surprising improvement in entrainment of worms lacking L-Cry under standard lab conditions sparked imaginative and well executed lunar phase shift experiments demonstrating that natural moonlight conditions (rather than lab moonlight which is rather more intense) improved phase-shifting/spawning synchrony in wildtype animals, more inline with that exhibited by *l-cry*^{-/-} animals under lab (harsh lighting) moonlight but less so in mutant animals. As such, and considering these observations, the authors proposed that L-CRY acts as 'gatekeeper' of light- 'blocking' the 'wrong' light from influencing the lunar clock and has a rather minor role as a photoreceptor. Then using biochemical *in vitro* approaches it was shown that L-CRY discriminates between sun and moonlight by forming photo-reduced states differently (in terms of duration and intensity of light required to induce reduction)- i.e. moonlight and sunlight states. These experiments seemed very well done and well controlled and may turn out to be really important in our fundamental understanding of how organisms can distinguish sunlight from moonlight to drive solar and lunar rhythms. A really stunning finding was that, *in vitro*, moonlight takes about the same time to induce photoreduction of the FAB chromophore (about 6h) as it does for *in vivo* lunar entrainment! Finally, the authors used immunochemical (immunohistochemistry and Western blotting) strategies to show light degradation dynamics of L-Cry and, intriguingly the differential nuclear and cytoplasmic localisation of L-Cry under moonlight (nuclear) and sunlight (cytoplasmic) conditions. Again, these data look convincing and well reported.

Together the data have enabled the authors to propose a tantalising and testable model whereby the intensity and duration differences of sunlight and moonlight (natural) are registered differently by L-Cry (biochemically and in its sub-cellular localisation) to influence lunar entrainment in this worm.

These findings are exciting; they are some of the very first to show how light signals may entrain a non-circadian phenotype and open a door into the circalunar clock- The model proposed is testable and needs following up. Whilst some of the discussion is speculative (and acknowledged as such) the paper is a certainly a significant advance in our understanding of the mechanistic basis of rhythmicity that is crucial to a hugely diverse organisms in the marine and terrestrial realms. As such the findings go beyond the chosen model worm and may have much wider impacts on our understanding of other circalunar organisms across a wide range of different taxa.

For the reasons given above, I think the manuscript should be published and have only minor (some very minor) queries/comments that might be considered:

We thank the reviewer for the detailed and accurate summary and positive assessment of our work.

1) Lines 195-197 state that according to the Mardia-Watson-Wheeler test there is less or NO significance in mutant worms before and after entrainment. I can't really see this in the figure/table. Perhaps the authors could elaborate or indicate in the main text which values indicate this?

We thank the reviewer for pointing this out. We rephrased the entire sentence and added the concrete reference to the relevant figure/table.

*Now p.7, lines 202-204: "For *l-cry*^{+/+} animals, the comparisons of the spawning distributions before and after re-entrainment show a 1000fold (months 1 versus C) and 10fold (months 1 versus D) higher statistical significance difference than the corresponding comparisons for *l-cry*^{-/-} worms (Fig.3f,g)."*

2) Line 202 'stronger'....this doesn't read too well- perhaps change to something like ' have shifted more robustly in response to.....or something like?

*Many thanks for the suggestion. We rephrased according to the reviewer's suggestion.
Now p.8, line 210.*

3) Line 220 ...'function itself IS a sensor....'

Rephrased. Now p.8, line 215

4) Figure 4d to 4d''' (not referred to in text). Comparing 4d to 4d''' (month 1 to month D) Unless, I am mistaken, month D has no significant deviation from uniformity ($p=0.07$). Can the authors say whether it is OK to consider this as re-entrainment in their analyses/interpretation if the shifted animals are not spawning at a particular phase of the lunar month?

Many thanks for making us aware that we should explain better why (and when) we consider it OK to talk of re-entrainment, despite the circular statistics showing $p=0.07$ for month D. In brief, we now explain more clearly that already month C is strong evidence for re-entrainment, because the nocturnal light stimuli themselves have absolutely no visible immediate effects on spawning time (see first and second re-entraining nocturnal light stimulus). The animals need to undergo massive physiological and behavioral changes for their nuptial dance, which need advance time for preparation (probably one of the reasons why an endogenous monthly timer is an evolutionary advantage). Based on an additional and extensive independent study on the acute effects of light on spawning behavior (see Zurl et al, <https://doi.org/10.1101/2021.04.16.440114>, already cited in the previous version of the manuscript), we also know that while the hour of spawning might change, the day of spawning (i.e. the monthly aspect) is clearly unaffected by the acute re-entraining nocturnal light. All this argues that the clear statistical significance of month C (in all tests) provides already strong evidence for re-entrainment. We added month D for completeness, but could remove it without any changes to our interpretation. We prefer to keep it, but now worded the text referring to it more precisely and carefully. Please see p.8-9, lines 232-245.

5) Line 394- Delete 'Already'. Or, say, "After only 10 minutes...."

Changed as requested, now p.13, line 380.

6) Line 404- delete 'already'.

Done. Now p.13, line 392.

7) Line 419- delete 'also'.

The section has been rephrased: now p.13, lines 405-410

8) Line 552- I suggest changing 'quick' to 'too hasty' or 'premature'.

Changed to „premature“, now .p18, line 585.

9) Lines 673-679. I didn't find this section very easy to follow. Was the supernatant column purified before PCR? Please clarify.

We now rephrased the entire paragraph to make it better understandable. See p.22, lines 706-717.

(The difficulty in understandability likely arose due to the technically different treatment of immature/premature versus mature worms. We now separated this as two different sections more visibly from each other to avoid confusion.)

10) Line 694- define PTW here, not later.

It is now defined at its first appearance (now line 732).

11) Line 732- 'slim' or 'skimmed' milk??

Corrected to „skim milk powder“. Thanks for pointing it out.

12) Line 777- anti-DIG or just DIG-UTP?

We thank the reviewer for pointing this out and rephrased to „Riboprobes were labelled with Digoxigenin-coupled to UTP (Roche Diagnostics)...“ (now p.25, lines 815-816)

13) The references have some formatting errors- such as species names lacking necessary formatting (italics etc). Please check.

Many thanks for pointing this out. All cross-checked, and if necessary corrected.

Reviewer #2 (Remarks to the Author):

The manuscript presents a study on the moonlight entrainment of the worm *Platynereis dumerilii*. This moonlight aspect of light entrainment is of high interest to the clock community, in particular with the molecular focus presented here. The photoreceptor cryptochrome is identified to govern the lunar clock and to help a differentiation between sun light and moon light levels. The experiments have been carefully conducted and demonstrate the importance of applying correct light sources with respect to both intensity and spectral distribution. The data are strong and support an assignment to cryptochrome. A further strength of the study is the evaluation of the effects of light on many different levels, from molecular data on purified protein to localization studies to behavioral assays on living knockout mutants. Serious concerns remain with respect to the interpretation, which is biophysically invalid. I additionally object to the novelty of 'the new concept that a light receptor does not just sense light, but by its intensity and duration can give light a valence'.

1. Abstract. The term 'valence interpreter' is nowhere explained in the manuscript. It is unnecessary to invent a new term for something available in the literature as 'high light' and 'low light' reception. This discrimination is commonly found in plant photoreception, for example for phototropin. Moreover, our eyes perform the same task by using rods and cones with different sensitivity to the same wavelength of light.

We thank the reviewer for pointing the aspect of the other high/low light sensors out to us. Indeed, our term "valence detector" was not well enough specified in the manuscript. We acknowledge that the distinction between 'high light' and 'low light' receptors is well established in the literature. We implemented these terms in our revised manuscript, in which we now also refer to other works on high/low light photoreceptors and thus embed our work better in this context. See especially p.16-17, lines 508-551 and our revised Figure 5j. The latter is now based on the different quantum yields of the two L-Cry monomers, as suggested by the reviewer in their comment 4 (see our response to comment 4).

The reason why we introduced the term "valence detector" is that we mean to refer to the temporal-ecological discrimination of light, i.e. that by its biochemical properties (translated into different cellular responses) L-Cry allows the organism to discriminate between the different temporal-ecological origins of light, i.e. sun versus moonlight, as well as moon phases. To our knowledge this is not present in the existing concepts on the function of the other mentioned photoreceptors. Even the supposedly "low" light receptors Cry2 of plants operate orders of magnitude away from moonlight, as their physiological function is to ensure reliable response across the sunlight intensity scale (also see discussion pp 14, lines 432-440, pp.16,17, lines 508-551).

*While animal rods can detect moonlight and cones the sunlight, their primary function is to ensure that visual images can be detected across the natural range of light, irrespective of the (natural) origin of light. Their function is **not** to discriminate if the light that is used for the visual process comes from the sun or the moon.*

Subsequent cognitive processes will then allow for a (conscious) discrimination between sun and moon, but this is due to the subsequent visual post-processing in the brain and not due to the intrinsic biochemical properties of the operating Opsins (at least not according to current knowledge).

In order to point out this novel aspect of the inherent biochemical discrimination properties of L-Cry under natural light conditions, we introduced the term "valence detector".

In order to make this viewpoint clearer, we

a) mention the term "valence interpreter" now only at the end of the abstract and put it directly in the ecological context of the natural light source interpretation: p.2, lines 40-41.

b) re-moved the term entirely from the results section and now introduce it in the context of the final summary scheme and the discussion section. This allows us to then directly discuss and explain the ecological light source interpretation aspect that we actually aim to point out. We hope that this better contextual embedding and position in the discussion clarifies our intention to discuss about the discrimination of natural light origins and moon phases for the orchestration of animal physiology and behavior.

Please see p.14, 15, lines 432-460 and Figure 7 legend: pp.31-32.

2. Line 72. The statement concerning previous work on the L-Cry is exaggerated. Reference 15 does not support 'that Pdu-L-Cry functions in principle as a light receptor in a heterologous cell expression system.' Please rephrase more carefully. Most evidence in ref. 15 is shown on tr-Cry and only very minor experiments have been performed on L-Cry. Moreover, the presence of tr-Cry should be included in the manuscript together with a differentiation in roles.

This is a misunderstanding of the reviewer. In reference 15 (Zantke et al , Cell Reports 2013) we made use of two well- established and accepted assays in the field of invertebrate Cryptochromes. These were established to provide a basic discrimination between two phylogenetically distinct groups of animal Cryptochromes by the lab of Steven Reppert, when the notion arose that Drosophila Cry is not the direct ortholog of mammalian Cry1 and Cry2. Instead, many insects possess an additional and different cry gene, that encodes a protein that is a direct ortholog to mammalian Cry1/2, which however got lost in Drosophila (Zhu, H et al Current Biology 2005).

Those two assays are based on known functions of the best characterized members of each of the two groups: *Drosophila* dCry and mammalian Cry1 and Cry2. dCry is characterized to act as a photoreceptor and upon light exposure to undergo (light-dependent) degradation. This feature has been extensively used in the tissue culture assays to test for possible light-sensitivity and signaling pathway similarities of dCry orthologs in other species. Similarly, mammalian Cry1 and Cry2 are characterized by their light-independent transcriptional repression function. Also for this feature there has been a cell culture assay established that is used to test for (light-independent) transcriptional repressor functions in a heterologous tissue culture set-up.

These assays are then used individually, or- if both types of animal Cryptochrome were identified-, these two assays have been used comparably to test for light-dependent (*Drosophila*-type Cry) degradation versus (mammalian-type Cry) transcriptional repression functions (e.g. Yuan, Q et al Mol. Biol. Evol. 2007, Öztürk, N et al JBC 2008, Rivera AS et al, JEB 2012, Zhang L et al, 2013, Curr Biol., Biscontin 2017, Scientific Reports).

In the cited work (ref. 15), we have followed this established approach, i.e. tested if Pdu-LCry and Pdu-tr-Cry show light dependent degradation or transcriptional repression of *bmal/clock*-mediated transcription under darkness in an heterologous (S2-cell) tissue culture assay.

Our work clearly showed that L-Cry gets light-dependently degraded, whereas Pdu-tr-Cry does NOT. In reverse, under darkness, Pdu-tr-Cry shows transcriptional repression function, whereas Pdu-L-Cry does NOT.

In contrary to the statement of the reviewer, there is simply no evidence that tr-Cry would function in a light-dependent fashion.

Of course, we are aware that these assays do not rule out a possible light-sensitivity of tr-Cry if tested in a different assay, nor did they show that the light-dependence of L-Cry has to be direct. However, they provided a reasonable base of judgement for us to focus our analyses on L-Cry as a putative light receptor. In order to convey this reasoning better, and to also help other readers who might not be familiar with the assays performed in ref. 15, we now provide a much-extended paragraph on the different Cryptochromes/photolyases, their phylogenetic relationships to known protein families, the mentioned established assays used to test animal cryptochromes, as well as our reasoning for the focus on L-Cry in the introduction. Please see p.3,4 lines 63-81.

On a final note, as described in the now extended paragraph, Platyneis, like many other invertebrates and lower vertebrates, possesses a Cryptochrome that is orthologous to the Cryptochromes found in plants, hence named plant-type (pt)-Crys. We did not focus on this Cry in our assays, as there is no information on expression nor light sensitivity in animals, and a genetic null mutant in the worms has no lunar phenotype (even not as double mutant with *l-cry-/-;pt-cry-/-*). The latter is not mentioned in the text, as we cannot refer to unpublished data. We also think that this would rather sidetrack the reader and goes beyond the scope of this manuscript, which is an in-depth characterization of L-Cry, and not on the comparisons of different worm Cryptochromes. We hope that these points make it much better understandable to the reviewer, why a focus on L-Cry as a putative light receptor involved in moonlight detection is a well justified choice.

3. Figure 5j. The scheme requires fundamental revisions.

In response to this comment as well as comments 4.) and 7.), we significantly revised former Figures 5j and EDF 6i. Furthermore, we exchanged (the revised) Figures 5j and Suppl.Fig 6i. Thereby the new main text Figure 5j only contains the information obtained from our UV/VIS spectra. These spectra only observe the FAD cofactor and not the protein and they do not capture short-lived intermediate states.

We also significantly changed the scheme now presented in Suppl.Fig 6i. It now includes information on proposed short-lived half-reduced intermediate states and possible allosteric effects. The reason for not only revising, but also switching the schemes is that the implementation of the very valuable comments of the reviewer resulted in a scheme that refers to possible protein conformations (now new Suppl.Fig 6i) with many different (speculative) possibilities. We felt that this could be too distracting in the main figure. The now main figure scheme 5j is much more based on our direct observations.

(Please also note that due to Nature Comm's formatting rules we had to rename all Extended Data Figure to Suppl.Fig.)

Sun light will also populate the moonlight state, only that the intensity is so high that it is further converted. The same applies to the recovery. Under moonlight conditions, the sunlight state will recover to the dark state via the moonlight state, only that the recovery from the moonlight state to the dark state is much faster and therefore the moonlight state does not accumulate.

As mentioned above, we extensively revised the former schemes Fig5j and Suppl.Fig 6i. We agree with the reviewer's argument that the interconversion between the fully oxidized dark state and the fully reduced sunlight state involves a short-lived half-reduced FADox FAD^{o-} intermediate state. However, the dark-state recovery of the sunlight state in presence of moonlight within minutes only occurs via a short-lived FAD^{o-} FADox intermediate (new Suppl.Fig 6i, left). This has to be different from the "true" moonlight state (new Suppl.Fig 6i, right), as the "true" moonlight state accumulates in presence of moonlight (moonlight maintains the moonlight state; "no moonlight recovery"). Therefore, the moonlight state to sunlight state transition (direct sunlight photoactivation) and the sunlight state to moonlight state transition (via the FADox FADox dark state) should follow distinct pathways, i.e. are not directly reversible (Suppl.Fig.6i).

Sunlight photoreduction and dark state recovery in darkness could go either via the “true” moonlight state or via the distinct short-lived FAD^{o-}- FADox intermediate (dashed orange (sunlight) and black (darkness) arrows). We now also implemented the reviewer comment 7.), i.e. that the fast decay of the half-reduced state in presence of moonlight “might be attributed to an allosteric regulation of the two homomers” into our revised model in Suppl.Fig 6i. Consequently (and in line with our arguments above), such proposed allosteric regulation leads to a half-reduced state, that is conformationally and kinetically different from the “true” moonlight state and therefore - in contrast to the moonlight state – does not accumulate under moonlight illumination. Furthermore, we propose that during dark state recovery of the sunlight state in darkness, when the different quantum yields of the two monomers do not play a role, the formation of transient mixed FAD^{o-}- FADox dimers with a significant lifetime is not mandatory, i.e. a transition via mixtures of “homogenous” FAD^{o-}- FAD^{o-}- dimers and FADox FADox dimers is also conceivable and in line with the observed absorbance spectra. Please note that the revised Suppl.Fig 6i includes the same set of solid arrows for the transitions between stably accumulating species as the revised Figure 5j. Suppl.Fig 6i additionally contains dashed arrows for pathways via transient intermediate states and indicates protein conformational changes as distinct shapes. The revised model including half-reduced transition states and proposed allosteric changes is described in the Figure legend of Suppl.Fig 6i, in the results on p.11-12 lines 333-360 and in the discussion on p. 16-18 lines 497-569.

4. L. 335. The idea is very reasonable to correlate the half-reduced state in the spectra with a homodimer with only one out of two flavins being converted. However, the difference between the two homomers is not the energy difference of the transition but the quantum yield. Sun light has the same energy as moon light when absorbed by FAD at 450 nm! (Revise extended data Fig. 6i accordingly) Only the number of photons is different. Therefore, one homomer has a high quantum yield, corresponding to a highly sensitive receptor. The second homomer has a low quantum yield, accordingly, many photons are required to convert the FAD to the anion radical. Differences in quantum yield of flavin are very common for LOV domains in phototropin or aureochrome and have been documented as well for bird and plant cryptochromes. Please do not involve redox potentials (l. 529), these are certainly a different issue.

We thank the reviewer for pointing this out to us. We revised the new Figure 5j (former EDF6i) according to the reviewer’s constructive suggestions. The revised Figure 5j now includes two relevant differences that address the reviewer’s comment: the different quantum yields of the flavins in the two L-Cry monomers (right) and the different photon numbers of moonlight and sunlight (left). Please see also results part p.11,12. In addition, we extended the discussion on p. 14, pp.16-18 to further discuss the different quantum yields of L-Cry monomers A and B, their implications for low/high light sensing along with a comparison to other photoreceptors, especially from plants. To illustrate the different photon numbers involved in the partial- and full photoreduction of L-Cry’s flavin by moonlight and sunlight, we also added a comparison of the photon numbers in the main absorption range of FADox and FAD^{o-} (330 – 510 nm), provided by 6 h and 12 h moonlight illumination and by 30 s to 5 min sunlight illumination (please see p. 12, lines 345-355, pp.16,17, lines 508-527). We also discuss the role of the different quantum yields in the two monomers for the formation of the moonlight state as an additional state, that enables L-Cry to discriminate between sunlight and moonlight and to recognize the full moon phase based on its duration. Please see: pp. 11,12 and pp. 14,16-18.

5. L. 325. The direct comparison of light levels in nature with those applied to a purified protein requires to take scattering and absorption by the tissue of the worm into account. How have these contributions been included in the setting of the light level?

While this might in principle be an interesting question, we ask the reviewer to consider two important points in the context of our Platynereis work:

1.) The L-Cry-positive cells are located directly under the cuticular surface. The cuticle in annelids consists of collagen, which is transparent and thin. It has been studied in detail in a close relative of Platynereis dumerilii, in Nereis virens (which also exhibits moon -controlled reproductive timing, see e.g. Bass, N., & Brafield, A. (1972). doi:10.1017/S0025315400021664, Bentley, M.G. et al, (1999). doi.org/10.1023/A:1017039110161). Nereis virens is even bigger than our experimental animal, yet its (collagen) cuticle is only 10um thick (Murray LW et al, (1981) doi: 10.1016/s0022-5320(81)80048-2). Also, the photoreceptors of the eyes of Platynereis lie directly underneath the same type of cuticle layer, which in this case is even thought to serve as a ‘cornea’ (Suschenko D and Purschke G, (2009) DOI 10.1007/s00435-008-0075-3.)

2.) In the Mediterranean Sea Platynereis dumerilii worms can be found in benthic areas spanning from few cm below the water surface to 10m. Obviously, several meters of sea water have a stronger impact on light intensity than 10um of transparent collagen. Given the habitat distribution, we decided to use light intensities that correspond to measurements at the intermediate depth of -4-5m below water surface.

Taking both points together, we think the impact of the overlying cuticle on light intensity is to our best judgement negligible for our study.

6. Figure 5a, line 289. The light scattering data (MALS) need to be converted into corresponding molecular weights. The panel does not reveal the experimental results. An example for how to analyze SEC-MALS results is shown in Winkler et al. (2013) Nat. Struct. Mol. Biol. 859 Figure 1b.

We thank the reviewer for pointing this out. We have corrected our MALS Figure 5a to depict the molecular weight/molar mass derived from the MALS experiment. We have revised Figure 5a and its legend as well as our statement about MALS in the results part, please see: p.10, lines 277-279.

7. L. 354. The fact that the moonlight state is not detected in the recovery does not allow for the conclusion that it is not formed in the recovery. It is simply not strongly populated which might be attributed to an allosteric regulation of the two homomers.

As we describe in detail in our response to the reviewer's comment #3.), we included an additional allosterically changed short-lived half-reduced intermediate state in our revised model in Suppl.Fig 6i (former Fig. 5j) (left species, depicted with a unique shape), that is different from the moonlight state. The possibility of allosteric regulation is included in the figure legend of Suppl.Fig. 6i, in the main text results on p. 11-12 and discussion on p. 18.

Directly in former Line 354 (now line 330), we replaced the word "entering" by "accumulating" the moonlight state in order to address the reviewers concern that L-Cry may also "enter" the moonlight state transiently upon sunlight- to dark-state recovery.

New sentence: "Hence, fully photoreduced sunlight-state L-Cry first has to return to the dark state before accumulating the moonlight state characterized by the stable presence of the partial FAD^{o-} product/FADox educt."

Minor points.

- Chemical compounds should be lower case not upper case throughout the manuscript.

Checked and changed.

- Replace 'adduct' with 'educt' throughout the manuscript.

Checked and changed.

- If all in vitro experiments were performed in the presence of the strong reductant TCEP, this fact needs to be included in the main text.

In addition to the original mentioning (p.20, line 639), this information is now also included in the results section (p. 10, line 281) and in the Materials and Methods section: p.21, line 665

- Extended data Figure 1d. The emission of the LED at 600 nm is zero not 10^{12} photons / (cm² s). A measurement needs to be included with the LED switched off to demonstrate the limit in sensitivity of the detector and to generate a zero line.

We thank the reviewer for pointing this aspect out to us. Actually, the measurement is correct, it is just depicted in logarithmic scale– which is normally not done for such light sources (especially not on the companies' datasheets!). The "conventional" plotting then often creates the impression that there should be no "background irradiation". For comparison, we now also present spectrum and intensity of this "blue-light dominated spectrum" plotted in linear scale. Please compare Suppl.Figs.1d and d', as well as to the Figure (in linear scale) on page 3 of datasheet of this LED (now added as Supplementary Material 1, Deep Blue: E001744 ACL01-SC-DDDD-E10-C01-L-L000 450). Please also check the manuscripts of other Cryptochrome papers in comparison. These papers typically either simply refer to the datasheets of the light source distributors, which either contains no spectra or the spectra are typically plotted in linear scale and relative (!) intensity (e.g. see Heinz U., Schlichting I., eLife 2016;5:e11860. DOI: 10.7554/eLife.11860; Krischer et al, Journal of Experimental Botany (2022), Vol. 73, No. 7 pp. 1934–1948, 2022 ; Juhas et al, FEBS J 2014 May;281(9):2299-311.doi: 10.1111/febs.12782. Alternatively, spectra are included in the manuscript, but again typically in linear plotting, e.g. Zoltowski et al, PNAS (2019), vol. 116 | no. 39 | 19449–19457.*

This unfortunately creates the impression that certain light sources have 0 emissions at other wavelengths, which is definitively not the case. This is why a re-measurement and plotting of absolute photon numbers in log scale is quite informative.

We now also added the information on the origin of this LED light to the methods section: p.21, lines 667-669, and corrected our phrasing in the manuscript to more appropriately refer to this light as a "blue-light dominated spectrum".

We are obviously aware that the light with the blue-light dominated spectrum is a rather artificial light. The reason why we had decided to include the data generated with it in our manuscript is for comparability reasons with other "blue lights" typically used for the characterization of Cryptochromes, such as Czarna et al (2013), doi: 10.1016/j.cell.2013.05.011 (which used exactly the same light source), but also others (e.g. see references above).

- Figures 5f and 5g. The right axis is corrupted and not at the same level as the left axis.

Corrected.

REVIEWERS' COMMENTS

Reviewer #1 (Remarks to the Author):

The authors have addressed my comments, made necessary and appropriate amendmnets to clairfy and tightened the text where needed.

I reccomend the manuscript for publication and congratulate the authors on their work.

Reviewer #2 (Remarks to the Author):

This work fundamentally improves our understanding of moon light reception and the adaptation of organisms to the lunar cycle.

The authors revised the manuscript in a very constructive manner and answered very well to all the issues raised. In particular, the model has been strongly revised and is now in agreement with biophysical laws. The aspect of a valence interpreter is now defined more carefully as appropriate in an ecological context. Accordingly, I fully support publication of the manuscript in its present state.